# Bridging Symmetry and Robustness: On the Role of Equivariance in Enhancing Adversarial Robustness

**Longwei Wang**[1], **Ifrat Ikhtear Uddin**[1], **KC Santosh**[1†],
**Chaowei Zhang**[2†], **Xiao Qin**[3], **Yang Zhou**[3†]

longwei.wang@usd.edu, ifratikhtear.uddin@coyotes.usd.edu, kc.santosh@usd.edu
cwzhang@yzu.edu.cn, xqin@auburn.edu, yangzhou@auburn.edu

[1]AI Research Lab, Department of Computer Science, University of South Dakota, USA
[2]School of Information and Artificial Intelligence, Yangzhou University, Yangzhou, China
[3]Department of Computer Science and Software Engineering, Auburn University, USA

## Abstract

Adversarial examples reveal critical vulnerabilities in deep neural networks by exploiting their sensitivity to imperceptible input perturbations. While adversarial training remains the predominant defense strategy, it often incurs significant computational cost and may compromise clean-data accuracy. In this work, we investigate an architectural approach to adversarial robustness by embedding group-equivariant convolutions—specifically, rotation- and scale-equivariant layers—into standard convolutional neural networks (CNNs). These layers encode symmetry priors that align model behavior with structured transformations in the input space, promoting smoother decision boundaries and greater resilience to adversarial attacks. We propose and evaluate two symmetry-aware architectures: a parallel design that processes standard and equivariant features independently before fusion, and a cascaded design that applies equivariant operations sequentially. Theoretically, we demonstrate that such models reduce hypothesis space complexity, regularize gradients, and yield tighter certified robustness bounds under the CLEVER (Cross Lipschitz Extreme Value for nEtwork Robustness) framework. Empirically, our models consistently improve adversarial robustness and generalization across CIFAR-10, CIFAR-100, and CIFAR-10C under both FGSM and PGD attacks, without requiring adversarial training. These findings underscore the potential of symmetry-enforcing architectures as efficient and principled alternatives to data augmentation-based defenses.

## 1 Introduction

Adversarial robustness, defined as the capacity of deep neural networks to produce consistent predictions under small and often imperceptible input perturbations, remains a fundamental and unresolved challenge in modern machine learning. Adversarial attacks exploit a model's sensitivity to small, norm-bounded input perturbations, leading to incorrect and often high-confidence predictions [1]. These perturbations typically exploit the model's reliance on spurious, non-semantic features that do not align with the true data-generating process [2]. A key contributing factor to this vulnerability is insufficient training data: when datasets are limited in size or diversity, models tend to overfit to superficial statistical patterns, such as background textures or local pixel correlations, rather than learning robust and generalizable representations [3, 4].

---

† Corresponding Authors.
Code available at: `https://github.com/ifratmitul/Role-of-Equivariance`

39th Conference on Neural Information Processing Systems (NeurIPS 2025).

Adversarial training has become a dominant approach to mitigate this vulnerability. It enhances model resilience by explicitly injecting adversarial examples into the training process, thereby guiding the model to focus on more discriminative, semantically grounded features [5, 6]. These adversarial examples expand the effective support of the training distribution, allowing models to develop wider decision margins and improved generalization to perturbed inputs [2, 7]. Despite its success, adversarial training is not without limitations: it is computationally expensive, may degrade performance on clean data, and is often specialized to the attack types seen during training. Moreover, it addresses robustness reactively by modifying data rather than proactively by redesigning the model architecture.

This motivates a fundamental question: *Can architectural priors alone improve adversarial robustness by encouraging models to align more closely with the geometric structure of data?* In this work, we explore this question through the lens of **equivariance**, the principle that model outputs should transform predictably under known input transformations. In particular, we investigate whether *embedding symmetry priors via group-equivariant convolutions* can enhance adversarial robustness in convolutional neural networks (CNNs) even in the absence of adversarial training.

Equivariance provides a principled mechanism for enforcing inductive biases that align with underlying symmetries in data. While standard CNNs are translation-equivariant by design, they are not inherently equivariant to other common transformations such as rotations and scalings. Group-equivariant convolutions generalize standard convolutions to be equivariant under larger transformation groups, such as the discrete rotation group $P4$ or scale groups [8–10]. These architectures encode transformation consistency directly into the weight-sharing scheme of the network, allowing the model to process rotated or rescaled inputs without relying on data augmentation. As a result, equivariant CNNs have demonstrated improved sample efficiency, stronger generalization, and greater interpretability across domains such as medical imaging, remote sensing, and physics-informed learning [11, 12].

Despite their success in structured learning tasks, the relationship between equivariance and adversarial robustness remains underexplored. Intuitively, adversarial perturbations often introduce changes that lie off the data manifold or violate known symmetries. By constraining the model to respond consistently along group-induced orbits and suppressing sensitivity to off-orbit perturbations, equivariant architectures may provide a natural defense mechanism. This raises a key research question: *How does architectural equivariance influence a model's resilience to adversarial perturbations, both theoretically and empirically?*

In this paper, we bridge the gap between symmetry enforcement and adversarial robustness by conducting a systematic study of CNN architectures that integrate standard, rotation-equivariant, and scale-equivariant convolutions. We propose two model designs to incorporate equivariant layers and evaluate their robustness properties across a spectrum of adversarial and natural corruption settings. Our main contributions are summarized as follows:

- We present a theoretical analysis demonstrating that equivariant architectures contract the hypothesis space, regularize gradient behavior, and admit tighter certified robustness bounds under the CLEVER (Cross Lipschitz Extreme Value for nEtwork Robustness) framework.

- We propose and compare two symmetry-aware CNN architectures *parallel* and *cascaded* that integrate standard, rotation-equivariant, and scale-equivariant convolutional layers. We show that the parallel design better preserves complementary feature spaces and achieves superior robustness. We explore two fusion strategies *simple concatenation* and *weighted summation* for combining features from multiple symmetry branches. Our findings indicate that concatenation consistently outperforms weighted fusion in adversarial settings.

- We validate our approach through extensive experiments on CIFAR-10, CIFAR-10C, and CIFAR-100 datasets, using FGSM and PGD attacks. Our results show that equivariant CNNs, particularly the parallel design with combined rotation and scale branches, significantly outperform standard CNNs in adversarial accuracy without requiring adversarial training.

## 2 Related Works

Trustworthy machine learning, which focuses on developing and deploying machine learning models that are not only accurate but also robust, private, fair, and explainable, has attracted active research in recent years [13–86]. Adversarial robustness, a core pillar of trustworthy ML, addresses the vul-

nerability of neural networks to imperceptible perturbations. In this work, we explore the intersection of symmetry-aware architectures and adversarial robustness, drawing from two key research areas.

## 2.1 Equivariant Neural Networks

Equivariance in neural networks ensures that transformations applied to input data lead to predictable and consistent transformations in the learned representations, aligning models with the inherent symmetries in the data. A landmark advancement in this field is the introduction of Group Equivariant Convolutional Networks (G-CNNs) [8], which extended traditional convolutional operations to group transformations, including rotations [87][88]. These networks demonstrated significant gains in performance and efficiency on symmetry-rich datasets such as MNIST and CIFAR-10. Subsequent progress led to the development of Harmonic Networks, which employed circular harmonics to achieve continuous rotational equivariance [9]. By eliminating the need for discrete approximations, these models improved the flexibility of equivariant frameworks [89]. Further extending these ideas, Scale-Equivariant Steerable Networks were introduced to address scale transformations, enabling the processing of multi-scale inputs without explicit data augmentation [90]. Steerable CNNs provided a versatile framework for handling equivariance under a range of transformations, paving the way for applications in domains such as medical imaging, astrophysics, and 3D object recognition [91].

More recently, spherical equivariance has garnered attention, with spherical CNNs being developed to handle data defined on spherical domains [12][92][93]. These models have found use in global-context tasks, including climate modeling and astrophysics [11]. Additionally, equivariant networks have been applied in molecular biology, utilizing molecular symmetries to predict chemical properties [94]. Despite these advancements, the susceptibility of equivariant models to adversarial perturbations remains an open area of investigation.

## 2.2 Adversarial Robustness

The discovery of adversarial examples exposed a significant limitation in neural networks, revealing their vulnerability to small, carefully designed perturbations [1]. These adversarial inputs exploit weaknesses in CNN architectures, leading to incorrect predictions. The Fast Gradient Sign Method (FGSM) formalized this issue as a single-step attack based on the direction of the loss gradient [95]. Later, Projected Gradient Descent (PGD) was introduced as a stronger iterative attack method, becoming a benchmark for adversarial testing [5]. Efforts to defend against such attacks have primarily focused on adversarial training, where models are trained using adversarially perturbed data to improve robustness [5][96][97]. However, this approach often results in reduced accuracy on clean data [2]. To mitigate these trade-offs, researchers have proposed architectural innovations, such as feature denoising modules [98] and preprocessing techniques like compression, resizing, and randomization [99][100][101], which aim to diminish the effect of adversarial perturbations.

Advanced defenses have also leveraged model uncertainty and interpretability. Randomized smoothing has emerged as a certified defense strategy [102][103][104], while ensemble methods have demonstrated improved robustness by combining multiple decision boundaries [105] [64]. Despite these promising developments, the potential to integrate the symmetry-preserving principles of equivariant networks into adversarial defense strategies remains largely untapped, leaving a valuable avenue for future research.

## 3 Equivariance and Adversarial Robustness

Neural networks are known to be vulnerable to *adversarial perturbations*: small, human-imperceptible modifications to input data that cause incorrect predictions with high confidence. This phenomenon is often attributed to the model's overreliance on non-semantic features and irregular decision boundaries. One principled approach to mitigating this sensitivity is to enforce architectural inductive biases aligned with known symmetries of the data distribution. Among such biases, *equivariance* has emerged as a theoretically grounded and empirically effective mechanism.

## 3.1 Equivariance in Neural Networks

**Definition 1** (Equivariant Function). *Let $G$ act on $\mathcal{X}$ via $T_g$ and on $\mathcal{Y} \subseteq \mathbb{R}^k$ via a representation $\rho(g) \in \mathrm{Aut}(\mathcal{Y})$. A function $f : \mathcal{X} \to \mathcal{Y}$ is said to be $G$-equivariant if:*

$$f(T_g x) = T_g f(x), \quad \forall g \in G,\ x \in \mathcal{X}.$$

Standard CNNs are translation-equivariant due to weight sharing across spatial positions. However, they lack equivariance to transformations like rotation and scaling. Group-equivariant CNNs (G-CNNs) generalize convolution to act equivariantly under more general groups $G$, such as $C_n$, $\mathrm{SO}(2)$, or dilation groups, thereby promoting symmetry-aligned representations.

Formally, let $G$ be a group with an associated action on the input space $\mathcal{X} \subset \mathbb{R}^d$, and let $\rho : G \to \mathrm{Aut}(\mathbb{R}^k)$ be a linear representation of $G$ acting on the feature space. A function $f : \mathcal{X} \to \mathbb{R}^k$ is said to be *equivariant* with respect to the group $G$ if it satisfies:

$$f(g \cdot x) = \rho(g)f(x), \quad \forall g \in G, \tag{1}$$

where $g \cdot x$ denotes the transformed input under the action of $g \in G$. Equivariance ensures that applying a transformation to the input leads to a predictable transformation in the output, thereby promoting stability and consistency in feature representations.

## 3.2 Adversarial Robustness and Margin Bounds

**Definition 2** (Adversarial Robustness). *A classifier $f : \mathbb{R}^d \to \mathbb{R}^k$ is said to be $(\varepsilon, p)$-robust at input $x \in \mathbb{R}^d$ if:*

$$f(x + \delta) = f(x), \quad \forall \delta \in \mathbb{R}^d,\ \|\delta\|_p \leq \varepsilon.$$

**Definition 3** (Margin Function). *Let $f_c(x)$ denote the logit score of the predicted class $c$, and $f_j(x)$ the score of class $j \neq c$. The class margin is:*

$$g_{c,j}(x) := f_c(x) - f_j(x).$$

**Definition 4** (Certified Robustness via Lipschitz Bound). *Let $g_{c,j}$ be locally Lipschitz with constant $L > 0$ near $x$. Then for any $\delta \in \mathbb{R}^d$ such that $\|\delta\|_p \leq \varepsilon$, we have:*

$$|g_{c,j}(x + \delta) - g_{c,j}(x)| \leq L\|\delta\|_p.$$

*Consequently, robustness against class $j$ is certified if:*

$$\varepsilon_{c \to j}^{(p)} \geq \frac{g_{c,j}(x)}{L}.$$

**Definition 5** (CLEVER Bound [106]). *Let $g_{c,j}$ be the margin function as above. The CLEVER bound estimates a certified perturbation radius as:*

$$\epsilon_{\min}^{(p)}(x) := \min_{j \neq c} \frac{g_{c,j}(x)}{L_q^{(j)}},$$

*where $1/p + 1/q = 1$, and*

$$L_q^{(j)} := \sup_{x' \in B_p(x,r)} \|\nabla g_{c,j}(x')\|_q$$

*is a data-dependent estimate of the local Lipschitz constant of $g_{c,j}$ in the dual norm.*

# 4 Theoretical Analysis of Adversarial Robustness with Equivariant Convolutions

This section presents a comprehensive theoretical framework for analyzing the adversarial robustness of group-equivariant neural networks. We develop the mathematical foundations necessary for understanding the relationship between equivariance and model sensitivity, formalize certified robustness bounds under Lipschitz constraints, and show how equivariant architectures induce smoother gradients and larger certified margins.

## 4.1 Mathematical Preliminaries and Equivariant Structures

**Definition 6** (Input Space and Model). *Let $\mathcal{X} \subset \mathbb{R}^d$ be the input space, and let $f : \mathcal{X} \to \mathbb{R}^k$ be a neural network mapping inputs to logit vectors. We assume $f$ is differentiable almost everywhere.*

**Definition 7** (Orbit and Quotient Space). *The orbit of a point $x \in \mathcal{X}$ under $G$ is the set $[x]_G := \{g \cdot x \mid g \in G\}$. The quotient space $\mathcal{X}/G$ is the collection of distinct orbits in $\mathcal{X}$.*

**Definition 8** (Jacobian and Lipschitz Constant). *If $f$ is differentiable at $x$, the Jacobian is $J_f(x) := \nabla f(x) \in \mathbb{R}^{k \times d}$, and the local Lipschitz constant is $L(x) := \|J_f(x)\|_2$.*

**Definition 9** (Adversarial Perturbation). *A vector $\delta \in \mathbb{R}^d$ is an adversarial perturbation at $x$ if $f(x + \delta) \neq f(x)$ and $\|\delta\|_p \leq \varepsilon$.*

## 4.2 Jacobian Structure and Lipschitz Regularity

**Definition 10** (Global Lipschitz Continuity). *A function $f$ is globally Lipschitz if there exists $L > 0$ such that:*
$$\|f(x_1) - f(x_2)\| \leq L \cdot \|x_1 - x_2\|, \quad \forall x_1, x_2 \in \mathbb{R}^d.$$

**Definition 11** (Jacobian under Equivariance). *If $f(g \cdot x) = \rho(g)f(x)$, then the Jacobian transforms as:*
$$J_f(g \cdot x) = \rho(g)J_f(x)Dg^{-1},$$
*where $Dg^{-1}$ is the Jacobian of the inverse transformation.*

**Lemma 1** (Jacobian Norm Invariance [107]). *If $\rho(g)$ and $Dg^{-1}$ are orthogonal matrices, then:*
$$\|J_f(g \cdot x)\|_2 = \|J_f(x)\|_2.$$

## 4.3 Certified Robustness via CLEVER Bounds

We analyze the certified adversarial robustness of group-equivariant networks through the CLEVER framework [106], which provides a lower bound on the minimum input perturbation required to induce misclassification. We show that equivariance yields gradient invariance over group orbits, leading to consistent and stronger robustness certification.

**Lemma 2** (Transformation of Margins under Group Equivariance [108]). *Let $f$ be $G$-equivariant, i.e.,*
$$f(g \cdot \mathbf{x}) = \rho(g)f(\mathbf{x}),$$
*where $\rho : G \to \mathrm{GL}(\mathbb{R}^k)$ is a linear representation. Then for any $j \neq c$,*
$$g_{c,j}(g \cdot \mathbf{x}) = \rho_{cc}(g)f_c(\mathbf{x}) - \rho_{jj}(g)f_j(\mathbf{x}).$$

**Lemma 3** (Gradient Transformation of Margin Function [107]). *Differentiating the margin function under the group action yields:*
$$\nabla g_{c,j}(g \cdot \mathbf{x}) = \rho(g)\nabla g_{c,j}(\mathbf{x})Dg^{-1},$$
*where $Dg^{-1} \in \mathbb{R}^{d \times d}$ is the Jacobian of the inverse group action.*

**Theorem 1** (Orbit-Invariance of Margin Gradient Norm). *If both $\rho(g)$ and $Dg^{-1}$ are norm-preserving (e.g., orthogonal matrices), then for all $g \in G$,*
$$\|\nabla g_{c,j}(g \cdot \mathbf{x})\|_q = \|\nabla g_{c,j}(\mathbf{x})\|_q.$$
*As a result, the Lipschitz constant of the margin function is invariant across the group orbit:*
$$L_q^{(j)} = \sup_{\mathbf{x}' \in B_p([\mathbf{x}]_G, r)} \|\nabla g_{c,j}(\mathbf{x}')\|_q,$$
*where $[\mathbf{x}]_G := \{g \cdot \mathbf{x} \mid g \in G\}$.*

The orbit-invariance of both the classification margin and its gradient norm establishes a compelling theoretical foundation for the robustness of group-equivariant networks. These models are not merely robust at isolated input points but offer uniform guarantees across entire equivalence classes of inputs linked by symmetry transformations. By construction, group-equivariant architectures preserve margin values consistently under group actions, ensuring that the discriminative separation between classes remains stable across symmetrically transformed instances. Furthermore, they inherently suppress gradient sensitivity in directions aligned with the symmetry structure of the data, effectively filtering out perturbations that respect these invariances. As a result, equivariant networks exhibit tighter and more reliable CLEVER-certified robustness bounds throughout the input space. Detailed proof is provided in Appendix A.1.

## 4.4 Equivariance-Induced Gradient Smoothing

A core source of adversarial vulnerability in neural networks is the irregularity of their input-output mappings, often reflected in sharp or non-smooth gradients. Group-equivariant networks mitigate this by imposing geometric constraints that smooth gradients over symmetric input transformations. The idea of this section is to analyze how group equivariance suppress adversarial vulnerability by smoothing the gradients of the network with respect to its inputs. Specifically, we show that group symmetries induce consistent, low-variance gradient fields along symmetric transformations, and reduce sensitivity to adversarial perturbations that deviate from these structured directions. A key source of adversarial fragility in neural networks stems from the irregularity of their input-output mappings, often manifesting as sharp or non-smooth gradients. Group-equivariant models mitigate this issue by embedding geometric constraints that align the model's behavior with inherent symmetries in the data, leading to smoother gradients and more stable decision boundaries.

**Definition 12** (Logit Gradient and Jacobian Matrix). *Let $f : \mathbb{R}^d \to \mathbb{R}^k$ be a differentiable classifier, and let $f_j(\mathbf{x})$ denote the logit for class $j$. The Jacobian of $f$ at $\mathbf{x}$ is:*

$$J_f(\mathbf{x}) := \nabla f(\mathbf{x}) = \begin{bmatrix} \nabla f_1(\mathbf{x})^\top \\ \vdots \\ \nabla f_k(\mathbf{x})^\top \end{bmatrix} \in \mathbb{R}^{k \times d},$$

*where $\nabla f_j(\mathbf{x}) \in \mathbb{R}^d$ denotes the gradient of the $j$-th logit with respect to the input. Each row of $J_f(\mathbf{x})$ characterizes how sensitive a particular output is to infinitesimal changes in different input directions.*

**Lemma 4** (Gradient Transformation under Group Equivariance [109]). *Let $f$ be a $G$-equivariant function, i.e.,*

$$f(g \cdot \mathbf{x}) = \rho(g)f(\mathbf{x}),$$

*with $\rho(g)$ a representation and $Dg^{-1}$ the Jacobian of the inverse group action. Then:*

$$\nabla f(g \cdot \mathbf{x}) = \rho(g) \cdot \nabla f(\mathbf{x}) \cdot Dg^{-1}.$$

**Definition 13** (Orbit-Averaged Gradient Field). *Define the per-logit gradient vector as $\phi_j(\mathbf{x}) := \nabla f_j(\mathbf{x})$. Then, the orbit-averaged gradient is:*

$$\overline{\phi}_j(\mathbf{x}) := \frac{1}{|G|} \sum_{g \in G} \nabla f_j(g \cdot \mathbf{x}) = \frac{1}{|G|} \sum_{g \in G} \rho(g) \nabla f_j(\mathbf{x}) Dg^{-1}.$$

**Lemma 5** (Smoothing via Orbit Averaging [108]). *Let $\boldsymbol{\delta} \in \mathbb{R}^d$ be a small perturbation. Then:*

$$\|\overline{\phi}_j(\mathbf{x}) - \phi_j(\mathbf{x})\| \ll \|\phi_j(\mathbf{x} + \boldsymbol{\delta}) - \phi_j(\mathbf{x})\|,$$

*especially when $\boldsymbol{\delta}$ is orthogonal to the group orbit $[\mathbf{x}]_G$. Thus, orbit-averaging suppresses high-frequency variations in the gradient field.*

**Theorem 2** (Directional Suppression of Off-Orbit Perturbations). *Let $f : \mathbb{R}^d \to \mathbb{R}^k$ be a differentiable function that is equivariant under the action of a compact group $G$, i.e.,*

$$f(g \cdot \mathbf{x}) = \rho(g)f(\mathbf{x}) \quad \text{for all } g \in G,$$

*where $\rho(g) \in \mathrm{GL}(\mathbb{R}^k)$ is a linear representation and $g \cdot \mathbf{x}$ is the group action on the input space. Suppose a perturbation vector $\boldsymbol{\delta} \in \mathbb{R}^d$ can be decomposed as:*

$$\boldsymbol{\delta} = \boldsymbol{\delta}_G + \boldsymbol{\delta}_\perp,$$

*where $\boldsymbol{\delta}_G \in T_\mathbf{x}([\mathbf{x}]_G)$ lies in the tangent space of the group orbit at $\mathbf{x}$, and $\boldsymbol{\delta}_\perp \perp T_\mathbf{x}([\mathbf{x}]_G)$. Then:*

$$\|\nabla f(\mathbf{x} + \boldsymbol{\delta}_\perp) - \nabla f(\mathbf{x})\|_2 \gg \|\nabla f(\mathbf{x} + \boldsymbol{\delta}_G) - \nabla f(\mathbf{x})\|_2.$$

*In particular, if $f$ is orbit-averaged over $G$, then $\|\nabla f(\mathbf{x} + \boldsymbol{\delta}_G) - \nabla f(\mathbf{x})\|_2 \to 0$, and off-orbit perturbations dominate gradient variability.*

This theorem formalizes the observation that equivariant networks inherently suppress gradient sensitivity along symmetry-respecting directions, while remaining susceptible to orthogonal, adversarial ones. This anisotropy in gradient variability contributes to improved adversarial robustness and smoother decision boundaries. Equivariance not only constrains functional outputs under transformations but also regularizes local geometry of the model's decision surface. Gradient smoothing and directional suppression enhance robustness by reducing sensitivity to adversarial perturbations, especially those orthogonal to the data manifold's symmetric structure. Detailed proof is provided in Appendix A.2.

## 4.5 Robustness Analysis of Scale Equivariance

Scale-equivariant neural networks do not satisfy the same assumptions required for certified robustness under isometric transformations such as rotations. Specifically, scale transformations alter the norm of the input, violating the orthogonality condition required in Lemma 1 and Theorem 1. Nevertheless, scale-equivariant architectures contribute to adversarial robustness through a different mechanism namely, gradient smoothing via multi-scale orbit averaging.

Let $x \in \mathbb{R}^d$ be an input and $G_s$ a finite group of scaling transformations. The orbit of $x$ under $G_s$ is defined as:
$$\mathcal{O}_s(x) = \{T_s(x) \mid s \in G_s\},$$
where $T_s(x) = s \cdot x$. We define the orbit-averaged gradient of a class logit function $\phi_j$ as:
$$\bar{\nabla}\phi_j(x) = \frac{1}{|G_s|} \sum_{s \in G_s} \nabla\phi_j(T_s x).$$

Unlike in the rotation-equivariant case, the norms $\|\nabla\phi_j(T_s x)\|$ are not preserved across the orbit, but the averaging process acts as a form of regularization. It reduces the gradient variance across local neighborhoods, thereby smoothing the decision boundary and dampening sensitivity to adversarial perturbations that rely on sharp gradients.

Scale-equivariant convolutional neural networks (CNNs) achieve robustness by explicitly enforcing equivariance across multiple spatial scales. This is typically done using scale-group convolutions, defined as:
$$[\Phi f](x) = \bigoplus_{s \in G_s} \psi_s * f(T_s^{-1}x),$$
where $\psi_s$ is the filter bank corresponding to scale $s$, $*$ denotes convolution, and $T_s(x) = s \cdot x$. The output is a scale-indexed feature map that captures the input structure across different resolutions.

This structure induces a smoothing effect both in the feature and gradient spaces. Specifically, consider the aggregated feature response at a given layer:
$$h(x) = \sum_{s \in G_s} w_s \cdot \phi_s(x), \quad \text{with } \phi_s(x) = \psi_s * f(T_s^{-1}x),$$

and its gradient with respect to the input:
$$\nabla h(x) = \sum_{s \in G_s} w_s \cdot \nabla\phi_s(x).$$

Although the gradient norms $\|\nabla\phi_s(x)\|$ scale with $s$, their aggregation smooths the overall gradient field by suppressing high-frequency components. This is analogous to a low-pass filter in the frequency domain and reduces the model's vulnerability to adversarial perturbations that exploit sharp local gradient changes. While scale-equivariant models fall outside the domain of certified robustness guarantees derived under norm-preserving assumptions, they introduce robustness via a complementary mechanism: smoothing the activation and gradient fields through multi-scale aggregation. This mechanism stabilizes the model's output under input perturbations and effectively regularizes its sensitivity, promoting robustness in practice.

## 5 Equivariance Enhanced Architectural Designs

### 5.1 Group Equivariant Convolutions

Group Equivariant Convolutional Networks (G-CNNs) generalize standard convolutional architectures by incorporating symmetry priors directly into the model design [8]. These networks are constructed to preserve equivariance under transformations defined by a group $G$, such as translations, rotations, or scalings. To achieve this, the conventional convolution operation is replaced by a *group convolution*, which aggregates features across group-transformed versions of both the input and the filters. In this work, we focus on two widely applicable instances: rotation-equivariant and scale-equivariant convolutions. Detailed formulations of these operations are provided in Appendix C.1.

## 5.2 Equivariance-Enforced Architectural Designs

We investigate two architectural strategies that integrate standard, rotation-equivariant, and scale-equivariant convolutions to enhance robust feature extraction: the *parallel* design and the *cascaded* design. Each approach offers a distinct balance between representational diversity and computational efficiency. Comprehensive architectural details are provided in Appendix C.2.

# 6 Experiments and Discussion

This section details the experimental setup used to evaluate the impact of adding rotation- and scale-equivariant convolutions on adversarial robustness and generalization. The experiments were conducted using three widely recognized datasets CIFAR-10, CIFAR-100, and CIFAR-10C to ensure a comprehensive evaluation of adversarial robustness, and generalization under natural corruptions. CIFAR-10C [110] is a variant of CIFAR-10 designed to evaluate corruption robustness. It includes 19 types of natural corruptions (e.g., Gaussian noise, motion blur, fog, and pixelation) applied at five levels of severity.

## 6.1 Models

To investigate the impact of equivariance on adversarial robustness, we designed and evaluated five CNN architectures with varying symmetry-aware modifications.

**Baseline Standard CNN** serves as the benchmark model, implemented with either 4 or 10 convolutional layers. **Parallel GCNN** replaces the first convolutional layer with two parallel branches: a standard convolution branch and a rotation-equivariant branch based on the discrete group P4. **Parallel GCNN with Rotation- and Scale-Equivariant Branches** extends the above by introducing a third scale-equivariant branch, enabling the model to process inputs across multiple geometric transformations. **Cascaded GCNN** adopts a sequential structure, where the input is first processed by a rotation-equivariant layer, followed by standard convolutions. **Weighted Parallel GCNN** uses the same three-branch structure as the previous parallel design but replaces feature concatenation with learnable fusion weights optimized during training.

## 6.2 Comparison of Adversarial Robustness under FGSM and PGD Attacks

We evaluated five models Baseline Standard CNN, Parallel GCNN, Parallel GCNN with Rotation- and Scale-Equivariant Branch, Cascaded GCNN, and Weighted Parallel GCNN on CIFAR-10 and CIFAR-100 datasets under FGSM and PGD attacks. To ensure a comprehensive understanding of the impact of network depth on robustness, we experimented with both 4-layer and 10-layer CNN architectures for all models.

In Figure 1, we present the adversarial robustness comparison of five models using 4-layer architectures on CIFAR-10 and CIFAR-100. The evaluation considers adversarial accuracies under FGSM and PGD attacks across a range of perturbation magnitudes ($\epsilon$). For CIFAR-10, the Parallel GCNN with Rotation and Scale Branch exhibited the highest robustness. The Parallel GCNN also performed well, but its robustness declined more rapidly compared to the combined model. On CIFAR-100, the overall robustness was lower due to increased class diversity and complexity. The Parallel GCNN with Rotation- and Scale-Equivariant Branch remained the most robust particularly at higher perturbations. The Parallel GCNN showed competitive performance at low perturbations but lagged behind the combined model, particularly at higher perturbations.

In Figure 2, we consider 10-layer architectures on CIFAR-10 and CIFAR-100 datasets. On both CIFAR-10 and CIFAR-100, the Parallel GCNN with Rotation and Scale-Equivariant Branch maintained the highest adversarial robustness across all perturbation levels. The Parallel GCNN with Rotation-Equivariant Branch also demonstrated strong robustness, though it was consistently outperformed by the combined model. Both the Cascaded GCNN and Weighted Parallel GCNN showed limited robustness, with adversarial accuracies dropping below 15% for FGSM and almost negligible for PGD attacks at higher perturbation levels.

To validate our theoretical framework under strict symmetry constraints, we evaluated fully equivariant architectures where all convolutional layers are equivariant, without any standard convolution branches. The 10-layer fully equivariant model achieves 73.01% FGSM and 64.96% PGD accuracy

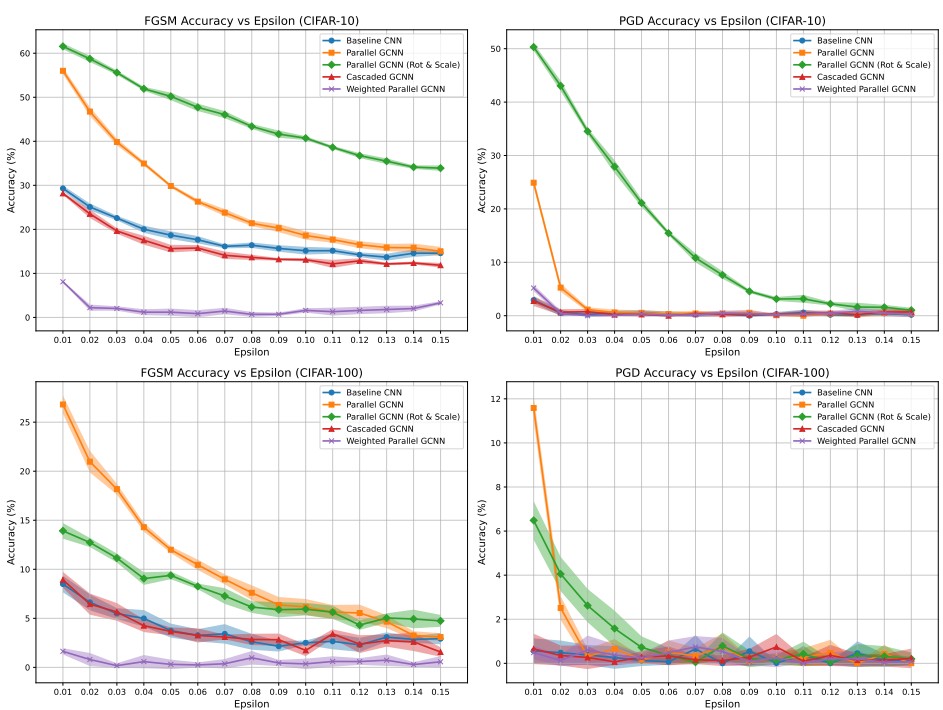

Figure 1: Adversarial robustness comparison of five models using 4-layer architectures on CIFAR-10 and CIFAR-100. Shaded regions represent ±1 standard deviation over 5 random seeds.

Table 1: Performance Analysis of Different Models under Various Corruption Types and Perturbation Levels on CIFAR10C (%)

| Corruption Type | BASELINE CNN | | | | CASCADED GCNN | | | | PARALLEL GCNN | | | | PARALLEL GCNN (R & S) | | | | WEIGHTED GCNN (R & S) | | | |
|---|---|---|---|---|---|---|---|---|---|---|---|---|---|---|---|---|---|---|---|---|
| | $\varepsilon_1$ | $\varepsilon_2$ | $\varepsilon_3$ | $\varepsilon_4$ | $\varepsilon_1$ | $\varepsilon_2$ | $\varepsilon_3$ | $\varepsilon_4$ | $\varepsilon_1$ | $\varepsilon_2$ | $\varepsilon_3$ | $\varepsilon_4$ | $\varepsilon_1$ | $\varepsilon_2$ | $\varepsilon_3$ | $\varepsilon_4$ | $\varepsilon_1$ | $\varepsilon_2$ | $\varepsilon_3$ | $\varepsilon_4$ |
| Brightness | 14.79 | 2.92 | 0.64 | 0.16 | 8.27 | 0.46 | 0.02 | 0.00 | **18.38** | 5.86 | 1.85 | 0.65 | 8.38 | 0.24 | 0.00 | 0.00 | 8.13 | 0.37 | 0.01 | 0.00 |
| Contrast | 3.25 | 0.71 | 0.39 | 0.33 | 1.31 | 0.00 | 0.00 | 0.00 | **7.35** | 2.52 | 0.46 | 0.05 | 0.69 | 0.00 | 0.00 | 0.00 | 0.59 | 0.00 | 0.00 | 0.00 |
| Defocus Blur | 7.92 | 1.33 | 0.32 | 0.11 | 3.09 | 0.04 | 0.00 | 0.00 | **13.15** | 3.22 | 0.79 | 0.20 | 2.72 | 0.03 | 0.00 | 0.00 | 2.80 | 0.04 | 0.00 | 0.00 |
| Elastic Transform | 6.80 | 1.11 | 0.28 | 0.07 | 1.90 | 0.03 | 0.00 | 0.00 | **11.46** | 2.68 | 0.66 | 0.19 | 1.92 | 0.01 | 0.00 | 0.00 | 2.02 | 0.03 | 0.00 | 0.00 |
| Fog | 3.64 | 0.45 | 0.12 | 0.03 | 1.31 | 0.01 | 0.00 | 0.00 | **6.97** | 1.24 | 0.21 | 0.04 | 1.39 | 0.00 | 0.00 | 0.00 | 1.11 | 0.00 | 0.00 | 0.00 |
| Frost | 9.32 | 1.45 | 0.30 | 0.10 | 3.15 | 0.17 | 0.01 | 0.00 | **11.90** | 3.32 | 1.02 | 0.38 | 3.44 | 0.10 | 0.01 | 0.00 | 3.07 | 0.15 | 0.01 | 0.00 |
| Gaussian Blur | 7.24 | 1.12 | 0.27 | 0.10 | 2.74 | 0.03 | 0.00 | 0.00 | **11.82** | 2.86 | 0.69 | 0.17 | 2.02 | 0.02 | 0.00 | 0.00 | 2.23 | 0.04 | 0.00 | 0.00 |
| Gaussian Noise | 5.38 | 1.02 | 0.34 | 0.14 | 1.82 | 0.03 | 0.00 | 0.00 | **14.68** | 2.66 | 0.53 | 0.12 | 1.84 | 0.00 | 0.00 | 0.00 | 1.78 | 0.01 | 0.00 | 0.00 |
| Glass Blur | 5.51 | 1.07 | 0.33 | 0.13 | 0.84 | 0.01 | 0.00 | 0.00 | **10.70** | 2.29 | 0.48 | 0.11 | 0.90 | 0.02 | 0.00 | 0.00 | 1.22 | 0.02 | 0.00 | 0.00 |
| Impulse Noise | 5.99 | 1.06 | 0.31 | 0.14 | 1.57 | 0.03 | 0.00 | 0.00 | **12.00** | 2.03 | 0.34 | 0.06 | 1.56 | 0.01 | 0.00 | 0.00 | 1.44 | 0.01 | 0.00 | 0.00 |
| JPEG Compression | 8.68 | 1.69 | 0.46 | 0.15 | 3.42 | 0.11 | 0.00 | 0.00 | **17.44** | 4.42 | 1.09 | 0.31 | 4.35 | 0.06 | 0.00 | 0.00 | 3.35 | 0.06 | 0.00 | 0.00 |
| Motion Blur | 6.34 | 0.91 | 0.23 | 0.07 | 2.11 | 0.02 | 0.00 | 0.00 | **10.46** | 2.68 | 0.69 | 0.19 | 1.82 | 0.02 | 0.00 | 0.00 | 1.84 | 0.03 | 0.00 | 0.00 |
| Pixelate | 9.04 | 1.66 | 0.40 | 0.14 | 4.45 | 0.14 | 0.00 | 0.00 | **17.05** | 4.43 | 1.08 | 0.28 | 5.70 | 0.11 | 0.00 | 0.00 | 4.69 | 0.09 | 0.00 | 0.00 |
| Saturate | 9.74 | 2.33 | 0.79 | 0.29 | 8.18 | 0.44 | 0.02 | 0.00 | **22.09** | 6.51 | 1.70 | 0.53 | 7.90 | 0.20 | 0.01 | 0.00 | 7.82 | 0.29 | 0.01 | 0.00 |
| Shot Noise | 5.62 | 1.07 | 0.38 | 0.17 | 2.36 | 0.05 | 0.00 | 0.00 | **16.03** | 3.09 | 0.61 | 0.13 | 2.71 | 0.01 | 0.00 | 0.00 | 2.33 | 0.03 | 0.00 | 0.00 |
| Snow | 9.71 | 1.85 | 0.52 | 0.19 | 3.71 | 0.22 | 0.02 | 0.00 | **15.06** | 3.95 | 1.23 | 0.42 | 4.08 | 0.14 | 0.02 | 0.00 | 3.93 | 0.19 | 0.03 | 0.01 |
| Spatter | 7.80 | 1.58 | 0.49 | 0.23 | 3.19 | 0.11 | 0.01 | 0.00 | **15.41** | 3.27 | 0.77 | 0.19 | 3.54 | 0.05 | 0.00 | 0.00 | 3.29 | 0.07 | 0.01 | 0.00 |
| Speckle Noise | 5.25 | 1.15 | 0.38 | 0.18 | 2.48 | 0.04 | 0.00 | 0.00 | **15.81** | 2.91 | 0.53 | 0.10 | 2.92 | 0.01 | 0.00 | 0.00 | 2.24 | 0.02 | 0.00 | 0.00 |
| Zoom Blur | 6.39 | 0.93 | 0.20 | 0.08 | 1.47 | 0.00 | 0.00 | 0.00 | **10.41** | 2.34 | 0.59 | 0.14 | 1.07 | 0.01 | 0.00 | 0.00 | 1.39 | 0.03 | 0.00 | 0.00 |
| Mean | 7.28 | 1.34 | 0.38 | 0.15 | 2.76 | 0.10 | 0.00 | 0.00 | **13.59** | 3.28 | 0.81 | 0.23 | 2.95 | 0.05 | 0.00 | 0.00 | 3.01 | 0.07 | 0.00 | 0.00 |
| Std | ±2.63 | ±0.58 | ±0.16 | ±0.07 | ±1.64 | ±0.13 | ±0.00 | ±0.00 | **±3.82** | ±1.29 | ±0.43 | ±0.16 | ±2.20 | ±0.07 | ±0.00 | ±0.00 | ±2.01 | ±0.10 | ±0.00 | ±0.00 |

[†] **R & S** represents Rotation and Scaling. $\varepsilon_i$ represents perturbation threshold where $i \in \{1, \ldots, 4\}$ with values $\{0.01, 0.02, 0.03, 0.04\}$ respectively.
**Bold**: Best performance across all models. Results averaged over 5 runs with standard deviation shown.

at $\varepsilon = 0.01$ on CIFAR-10, confirming that orbit-invariant gradient regularization compounds beneficially when symmetry is enforced end-to-end. Complete results for fully equivariant architectures are provided in Appendix D.

Our equivariant models achieve these robustness improvements without adversarial training. For context, we compare our 10-layer equivariant model against a standard CNN trained with PGD adversarial training in Appendix E.

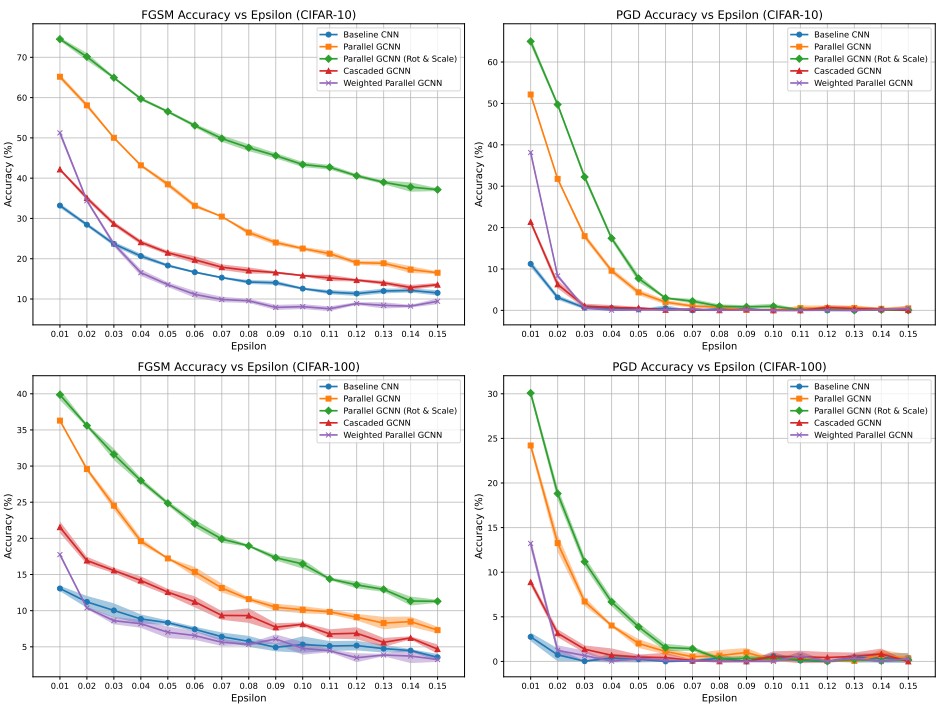

Figure 2: Adversarial robustness comparison of five models using 10-layer architectures on CIFAR-10 and CIFAR-100. Shaded regions represent ±1 standard deviation over 5 random seeds.

Table 1 presents the performance analysis of various models on CIFAR-10C under a range of corruption types and perturbation levels. The Parallel GCNN model consistently achieved the best performance across most corruption types, especially for lower perturbation thresholds. Parallel GCNN with Rotation and Scale Equivariance does not perform so well compared with baseline model under the data corruption. We assess the robustness of the proposed Parallel GCNN models using the maximum invariant perturbation metric [111], which quantifies the largest input perturbations a model can tolerate without altering the model's prediction. To further understand the contribution of each equivariant convolutional module, we perform ablation studies under default settings. The experimental details and visualizations of these studies are provided in Appendix F.

## 7 Conclusion

In this work, we conducted a systematic investigation into the role of architectural symmetry enforcement in improving adversarial robustness. By incorporating rotation- and scale-equivariant convolutions into standard CNNs, we demonstrated that symmetry-aware models could achieve improved resilience against adversarial attacks without relying on adversarial training or extensive data augmentation. Our theoretical analysis showed that equivariant architectures reduced hypothesis space complexity, regularized gradient behavior, and yielded tighter CLEVER-certified robustness bounds. These models consistently preserved classification margins under group transformations and suppressed gradient sensitivity in directions aligned with the data manifold's symmetry structure. Future work could extend these insights to larger-scale datasets, broader threat models, and more expressive network architectures.

## Acknowledgment

This work was supported by the National Science Foundation under Grant No. #2346643, and OAC-2313191, the U.S. Department of Defense under Award No. #FA9550-23-1-0495, the U.S. Department of Education under Grant No. P116Z240151 and National Research Platform (NRP) Nautilus HPC cluster [112]. Any opinions, findings, conclusions or recommendations expressed in this material are those of the author(s) and do not necessarily reflect the views of the National Science Foundation, the U.S. Department of Defense, or the U.S. Department of Education.

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

# A Technical Appendices and Supplementary Material

## A.1 Proof of Theorem 1

*Proof.* We begin by computing the gradient of the margin function $g_{c,j}$ at the transformed input $g \cdot \mathbf{x}$. Given the equivariance property:

$$f(g \cdot \mathbf{x}) = \rho(g)f(\mathbf{x}),$$

we write the individual logits as:

$$f_i(g \cdot \mathbf{x}) = [\rho(g)f(\mathbf{x})]_i = \sum_{m=1}^{k} \rho_{im}(g)f_m(\mathbf{x}).$$

Assuming $\rho(g)$ is diagonal (as in classification tasks where each logit transforms independently), the above reduces to:

$$f_i(g \cdot \mathbf{x}) = \rho_{ii}(g)f_i(\mathbf{x}).$$

Thus, the margin function transforms as:

$$g_{c,j}(g \cdot \mathbf{x}) = f_c(g \cdot \mathbf{x}) - f_j(g \cdot \mathbf{x}) = \rho_{cc}(g)f_c(\mathbf{x}) - \rho_{jj}(g)f_j(\mathbf{x}).$$

Now compute the gradient using the chain rule. Since $g \cdot \mathbf{x}$ is a diffeomorphism and $f$ is differentiable, we have:

$$\nabla g_{c,j}(g \cdot \mathbf{x}) = \nabla_{\mathbf{z}} g_{c,j}(\mathbf{z})\big|_{\mathbf{z}=g \cdot \mathbf{x}} = \nabla g_{c,j}(g \cdot \mathbf{x}) = J_{g_{c,j}}(g \cdot \mathbf{x}) = \nabla\left(f_c(g \cdot \mathbf{x}) - f_j(g \cdot \mathbf{x})\right).$$

Applying the chain rule to each term:

$$\nabla f_c(g \cdot \mathbf{x}) = \nabla\left(\rho_{cc}(g)f_c(\mathbf{x})\right) = \rho_{cc}(g) \cdot \nabla f_c(\mathbf{x}) \cdot Dg^{-1},$$

$$\nabla f_j(g \cdot \mathbf{x}) = \rho_{jj}(g) \cdot \nabla f_j(\mathbf{x}) \cdot Dg^{-1}.$$

Hence:

$$\begin{aligned}
\nabla g_{c,j}(g \cdot \mathbf{x}) &= \nabla f_c(g \cdot \mathbf{x}) - \nabla f_j(g \cdot \mathbf{x}) \\
&= \rho_{cc}(g)\nabla f_c(\mathbf{x})Dg^{-1} - \rho_{jj}(g)\nabla f_j(\mathbf{x})Dg^{-1} \\
&= \left(\rho_{cc}(g)\nabla f_c(\mathbf{x}) - \rho_{jj}(g)\nabla f_j(\mathbf{x})\right)Dg^{-1}.
\end{aligned}$$

We now compute the $\ell_q$-norm:

$$\|\nabla g_{c,j}(g \cdot \mathbf{x})\|_q = \left\|\left(\rho_{cc}(g)\nabla f_c(\mathbf{x}) - \rho_{jj}(g)\nabla f_j(\mathbf{x})\right)Dg^{-1}\right\|_q.$$

Assume that $\rho_{cc}(g), \rho_{jj}(g) \in \{+1, -1\}$ and that $Dg^{-1}$ is orthogonal:

$$\|Dg^{-1}\mathbf{v}\|_q = \|\mathbf{v}\|_q, \quad \forall \mathbf{v} \in \mathbb{R}^d.$$

Then:

$$\begin{aligned}
\|\nabla g_{c,j}(g \cdot \mathbf{x})\|_q &= \left\|\left(\rho_{cc}(g)\nabla f_c(\mathbf{x}) - \rho_{jj}(g)\nabla f_j(\mathbf{x})\right)Dg^{-1}\right\|_q \\
&= \|\rho_{cc}(g)\nabla f_c(\mathbf{x}) - \rho_{jj}(g)\nabla f_j(\mathbf{x})\|_q \\
&= \|\nabla g_{c,j}(\mathbf{x})\|_q.
\end{aligned}$$

Therefore, the norm of the gradient of the margin function is invariant under the group action:

$$\|\nabla g_{c,j}(g \cdot \mathbf{x})\|_q = \|\nabla g_{c,j}(\mathbf{x})\|_q, \quad \forall g \in G.$$

Since this holds for every $g \in G$, the Lipschitz constant of $g_{c,j}$ over the group orbit $[\mathbf{x}]_G$ satisfies:

$$L_q^{(j)} = \sup_{\mathbf{x}' \in B_p([\mathbf{x}]_G, r)} \|\nabla g_{c,j}(\mathbf{x}')\|_q = \|\nabla g_{c,j}(\mathbf{x})\|_q.$$

This completes the proof. $\qquad\square$

## A.2 Proof of Theorem 2

*Proof.* The key idea is that group equivariance induces structured regularity over the orbit of the input space. For any $g \in G$, by applying the chain rule to the equivariant condition $f(g \cdot \mathbf{x}) = \rho(g)f(\mathbf{x})$, we obtain the Jacobian transformation law:

$$J_f(g \cdot \mathbf{x}) = \rho(g)J_f(\mathbf{x})Dg^{-1}, \tag{2}$$

where $Dg^{-1} \in \mathbb{R}^{d \times d}$ denotes the Jacobian of the inverse group action $g^{-1} \cdot \mathbf{x}$. Because both $\rho(g)$ and $Dg^{-1}$ are orthogonal, the Jacobian norm is preserved:

$$\|J_f(g \cdot \mathbf{x})\|_2 = \|J_f(\mathbf{x})\|_2. \tag{3}$$

Now define the orbit-averaged gradient for each class logit $f_j(\mathbf{x})$ as:

$$\overline{\nabla f_j}(\mathbf{x}) := \frac{1}{|G|} \sum_{g \in G} \nabla f_j(g \cdot \mathbf{x}). \tag{4}$$

Using Eq. (2), we get:

$$\overline{\nabla f_j}(\mathbf{x}) = \frac{1}{|G|} \sum_{g \in G} \rho(g)\nabla f_j(\mathbf{x})Dg^{-1}. \tag{5}$$

This is a linear averaging operator acting on $\nabla f_j(\mathbf{x})$, which has the effect of projecting the gradient onto the invariant subspace under the group. High-frequency components in directions orthogonal to $T_{\mathbf{x}}([\mathbf{x}]_G)$ tend to cancel due to the group symmetries, while components aligned with the orbit are reinforced.

We now analyze the gradient difference along perturbations in each direction.

**Case 1:** Perturbation $\boldsymbol{\delta}_G \in T_{\mathbf{x}}([\mathbf{x}]_G)$:

Let $\mathbf{x}' = \mathbf{x} + \boldsymbol{\delta}_G$. Since $\boldsymbol{\delta}_G$ lies tangent to the orbit, we have $\mathbf{x}' \in [\mathbf{x}]_G$, and hence by the equivariance property and Eq. (3),

$$\nabla f(\mathbf{x}') \approx \nabla f(\mathbf{x}),$$

and thus,

$$\|\nabla f(\mathbf{x} + \boldsymbol{\delta}_G) - \nabla f(\mathbf{x})\|_2 \approx 0.$$

**Case 2:** Perturbation $\boldsymbol{\delta}_\perp \perp T_{\mathbf{x}}([\mathbf{x}]_G)$:

Let $\mathbf{x}'' = \mathbf{x} + \boldsymbol{\delta}_\perp$. Because this point lies **off the symmetry manifold**, the equivariant structure offers no constraint on the variation of the gradient. Consequently,

$$\|\nabla f(\mathbf{x} + \boldsymbol{\delta}_\perp) - \nabla f(\mathbf{x})\|_2$$

is not bounded by symmetry and may vary substantially—especially in directions where $\nabla f$ contains high-frequency components not captured by the group action.

Moreover, the difference between smoothed and unsmoothed gradients satisfies:

$$\left\|\overline{\nabla f_j}(\mathbf{x}) - \nabla f_j(\mathbf{x})\right\|_2 \ll \|\nabla f_j(\mathbf{x} + \boldsymbol{\delta}_\perp) - \nabla f_j(\mathbf{x})\|_2. \tag{6}$$

This shows that the orbit-averaged gradient stabilizes variation, while off-orbit perturbations significantly alter the local gradient field.

Combining Eqs. (A.2) and (6), we conclude:

$$\|\nabla f(\mathbf{x} + \boldsymbol{\delta}_\perp) - \nabla f(\mathbf{x})\|_2 \gg \|\nabla f(\mathbf{x} + \boldsymbol{\delta}_G) - \nabla f(\mathbf{x})\|_2,$$

which proves the theorem. $\qquad \square$

# B  Limitations

Despite its theoretical and empirical strengths, this work has several limitations. First, the theoretical framework focuses solely on local Lipschitz bounds (e.g., CLEVER), which may not capture broader robustness phenomena such as margin distributions or adversarial risk. Second, empirical evaluations are limited to small-scale datasets and common $\ell_p$-norm attacks, leaving open questions about scalability and robustness to more diverse threat models. Finally, the integration of equivariant layers is confined to early stages of the network, and the computational trade-offs of group convolutions remain unquantified.

# C  Equivariance Enhanced Architectural Designs

## C.1  Group Equivariant Convolutions

Group Equivariant Convolutional Networks (G-CNNs) extend the conventional convolutional framework by embedding symmetry priors directly into the architecture [8]. These models are designed to maintain equivariance under transformations defined by a group $G$, such as rotations, translations, or scalings.

To implement such equivariance in convolutional neural networks, the standard convolution operation is replaced with a *group convolution* that aggregates information over the transformed versions of both input and filters. We now instantiate this framework for two practically important cases: rotation and scale transformations.

### C.1.1  Rotation-Equivariant Convolutions

To embed rotational symmetry into the network, we consider a discrete rotation group $G = P4$, which consists of four planar rotations: $0°, 90°, 180°$, and $270°$. In this setting, group convolution is defined as:

$$[f * g](u) = \sum_{v \in G} f(v^{-1}u)g(v), \tag{7}$$

where $f$ is the input feature map, $g(v)$ is the group-transformed filter corresponding to transformation $v \in G$, and $u$ denotes the spatial coordinate. This formulation guarantees that the resulting feature map transforms predictably under group actions:

$$[f * g](g \cdot u) = \rho(g)[f * g](u), \quad \forall g \in G, \tag{8}$$

thereby satisfying the equivariance condition in Equation (1). By construction, these layers promote feature consistency across rotated versions of the input, enhancing robustness and generalization.

### C.1.2  Scale-Equivariant Convolutions

Scale transformations are another common form of variability in visual data, especially when objects appear at different sizes or distances. To build scale-equivariant architectures, we define a discrete scale group $G_s = \{\alpha_1, \alpha_2, \ldots, \alpha_k\} \subset \mathbb{R}^+$, which acts on the input space via isotropic resizing:

$$\alpha \cdot x := \text{Resize}(x, \alpha), \quad \alpha \in G_s, \tag{9}$$

where $\text{Resize}(x, \alpha)$ uniformly scales the spatial dimensions of the input $x$ by a factor $\alpha$, typically using bicubic interpolation. Each scaled input is then processed independently using a shared convolutional function $f_{\text{scale}}$, resulting in multiple feature maps:

$$F_{\text{scale},i} = f_{\text{scale}}(\alpha_i \cdot x), \quad i = 1, \ldots, k. \tag{10}$$

These outputs are then resized back to a common resolution and combined via an aggregation operation, such as channel-wise concatenation or averaging:

$$F_{\text{scale}}(x) = \text{Aggregate}(F_{\text{scale},1}, \ldots, F_{\text{scale},k}). \tag{11}$$

The resulting representation is equivariant to the scale group $G_s$, satisfying:

$$F_{\text{scale}}(\beta \cdot x) = \rho(\beta)F_{\text{scale}}(x), \quad \forall \beta \in G_s, \tag{12}$$

where $\rho(\beta)$ is the corresponding transformation (e.g., channel permutation or spatial rescaling) applied to the feature space. This construction allows the network to maintain consistent semantic representations across a range of object sizes, improving its ability to generalize under scale variation.

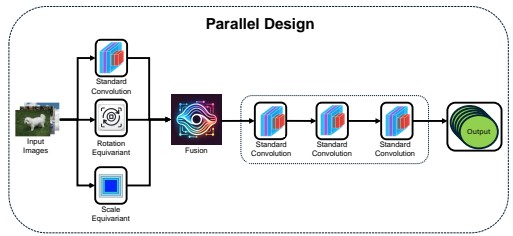

Figure 3: Parallel design of the CNN architecture. Input is processed through standard, rotation-equivariant, and scale-equivariant branches independently. The resulting features are subsequently fused to form a unified representation.

Figure 4: Cascaded design of the CNN architecture. Convolutional layers are applied sequentially: standard, then rotation-equivariant, followed by scale-equivariant.

Both rotation- and scale-equivariant convolutions are special cases of the general group-equivariant convolutional framework. They embed structured inductive biases into the network that enable efficient learning and enhanced generalization.

## C.2 Equivariance Enforced Architectural Designs

We explore two architectural strategies that combine standard, rotation-equivariant, and scale-equivariant convolutions for robust feature extraction: the *parallel* design and the *cascaded* design. Each strategy targets a different trade-off between representation diversity and processing efficiency.

## C.3 Parallel Design

In the parallel design, the input is processed simultaneously through three branches: a standard convolutional branch for translational equivariant features, a rotation-equivariant branch that encodes orientation equivariance via group convolutions, and a scale-equivariant branch that captures multi-scale features.

Let $\mathbf{x} \in \mathbb{R}^{C \times H \times W}$ denote the input image, where $C$, $H$, and $W$ are the number of channels and the spatial height and width, respectively. The outputs from each branch are given by:

$$\mathbf{F}_{\text{standard}} = f_{\text{standard}}(\mathbf{x}), \quad \mathbf{F}_{\text{rot}} = f_{\text{rot}}(\mathbf{x}), \quad \mathbf{F}_{\text{scale}} = f_{\text{scale}}(\mathbf{x}), \tag{13}$$

where $f_{\text{standard}}$, $f_{\text{rot}}$, and $f_{\text{scale}}$ are the respective processing functions. The final representation is obtained by fusing the branch outputs:

$$\mathbf{F}_{\text{fused}} = g(\mathbf{F}_{\text{standard}}, \mathbf{F}_{\text{rot}}, \mathbf{F}_{\text{scale}}), \tag{14}$$

where $g(\cdot)$ denotes the fusion strategy. Figure 3 illustrates this design, emphasizing the independence of the branches and their contribution to robust feature diversity.

## C.4 Cascaded Design

In contrast to the parallel strategy, the cascaded design applies equivariant operations sequentially. The input first passes through a standard convolutional layer, followed by rotation-equivariant processing, and finally scale-equivariant processing:

$$\mathbf{F}_{\text{rot}} = f_{\text{rot}}(f_{\text{standard}}(\mathbf{x})), \quad \mathbf{F}_{\text{scale}} = f_{\text{scale}}(\mathbf{F}_{\text{rot}}), \tag{15}$$

resulting in the final representation:

$$\mathbf{F}_{\text{fused}} = g(\mathbf{F}_{\text{scale}}). \tag{16}$$

This design simplifies the model by reusing intermediate representations. However, it may also introduce feature redundancy or suppress early-layer diversity, as subsequent operations act on already transformed representations. Figure 4 depicts the cascaded architecture, highlighting the flow of information through progressively specialized layers.

Table 2: Adversarial robustness of fully equivariant architectures on CIFAR-10 (%). Deeper models benefit from compounding gradient regularization when equivariance is enforced end-to-end.

| Architecture | $\varepsilon = 0.01$ | | $\varepsilon = 0.03$ | | $\varepsilon = 0.05$ | | $\varepsilon = 0.10$ | |
|---|---|---|---|---|---|---|---|---|
| | FGSM | PGD | FGSM | PGD | FGSM | PGD | FGSM | PGD |
| 4-Layer Equivariant | 65.65 | 52.20 | 53.78 | 23.30 | 47.08 | 15.85 | 37.95 | 7.01 |
| 10-Layer Equivariant | **73.01** | **64.96** | **67.09** | **52.37** | **60.23** | **37.80** | **44.93** | **12.46** |

Table 3: Adversarial robustness of fully equivariant architectures on CIFAR-100 (%). Consistent improvements across depth validate that orbit-invariant gradient regularization scales effectively.

| Architecture | $\varepsilon = 0.01$ | | $\varepsilon = 0.03$ | | $\varepsilon = 0.05$ | | $\varepsilon = 0.10$ | |
|---|---|---|---|---|---|---|---|---|
| | FGSM | PGD | FGSM | PGD | FGSM | PGD | FGSM | PGD |
| 4-Layer Equivariant | 38.40 | 21.59 | 26.63 | 8.92 | 22.06 | 5.66 | 15.96 | 2.39 |
| 10-Layer Equivariant | **50.60** | **36.29** | **42.02** | **21.34** | **36.09** | **12.01** | **24.68** | **4.08** |

# D   Fully Equivariant Architectures

To validate that our theoretical predictions hold for end-to-end equivariant models, we evaluate architectures where *all* convolutional layers enforce rotation equivariance under the P4 group (4 discrete rotations). These models are constructed by sequentially stacking rotation-equivariant blocks, with each block comprising an equivariant convolution, group-aware batch normalization, ReLU activation, and max pooling.

Tables 2 and 3 demonstrate that fully equivariant models achieve substantial adversarial robustness without adversarial training. The 10-layer architecture consistently outperforms the 4-layer variant across all perturbation levels, with improvements of 7-15% in FGSM accuracy and 10-30% in PGD accuracy at moderate perturbations. This depth-dependent improvement validates our theoretical prediction: when symmetry constraints are enforced throughout the network, orbit-invariant gradient regularization compounds beneficially across layers. This contrasts sharply with standard CNNs, where increased depth often amplifies adversarial vulnerability. The results confirm that equivariance must be treated as a network-wide architectural principle rather than localized feature extraction, demonstrating that certified robustness guarantees translate into measurable improvements under practical gradient-based attacks.

# E   Comparison with Adversarial Training

To contextualize the robustness improvements from architectural symmetry enforcement, we compare our approach against standard adversarial training. Table 4 presents results on CIFAR-10, where a standard CNN trained with PGD-based adversarial training is compared against our 10-layer fully equivariant G-CNN trained without any adversarial examples.

Our equivariant model achieves superior FGSM robustness and maintains competitive PGD robustness despite not being trained on adversarial examples. This demonstrates that symmetry-aware architectures provide intrinsic robustness through geometric constraints alone, offering a complementary and potentially more efficient alternative to data augmentation-based defenses. The slight advantage in FGSM accuracy and competitive PGD performance suggest that equivariant priors effectively regularize the decision boundary without the computational overhead of adversarial training.

Table 4: Comparison with Adversarial Training on CIFAR-10.

| Model | $\varepsilon = 0.01$ | | $\varepsilon = 0.02$ | | $\varepsilon = 0.03$ | | $\varepsilon = 0.04$ | | $\varepsilon = 0.05$ | |
|---|---|---|---|---|---|---|---|---|---|---|
| | FGSM | PGD | FGSM | PGD | FGSM | PGD | FGSM | PGD | FGSM | PGD |
| CNN + AT | 74.5 | 67.0 | 70.2 | 60.4 | 66.1 | 54.0 | 61.7 | 48.3 | 57.3 | 42.1 |
| G-CNN (no AT) | 73.01 | 64.96 | 70.16 | 58.87 | **67.09** | 52.37 | **63.77** | 45.52 | **60.23** | 37.80 |

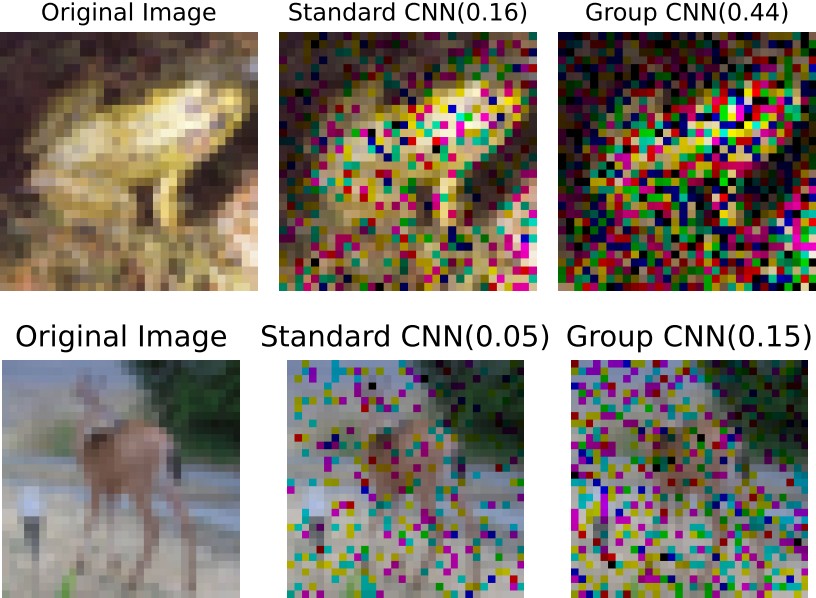

Figure 5: Visualization of perturbation tolerance for the Baseline CNN and the Parallel GCNN. The top row shows the original image (left), followed by the maximum perturbation ($\epsilon = 0.16$) for the Baseline CNN (middle), and the maximum perturbation ($\epsilon = 0.44$) for the Parallel GCNN (right). Similarly, the bottom row shows the original image (left), with the maximum perturbation ($\epsilon = 0.05$) for the Baseline CNN (middle), and the maximum perturbation ($\epsilon = 0.15$) for the Parallel GCNN (right).

## F Perturbation Tolerance Visualization and Ablation Study

### F.1 Visualization

We evaluated the robustness of the Parallel GCNN models using the metric of *maximum invariant perturbation* [111]. This metric quantifies the highest level of perturbations that a model can withstand without significant degradation in performance. The visualization in the figure compares the perturbation levels tolerated by the baseline CNN and the Parallel GCNN . As shown in the figure 5, the Parallel GCNN model demonstrates a substantially higher tolerance for perturbations compared to the baseline CNN. For example, the baseline CNN can tolerate a perturbation rate of $\epsilon = 0.05$, whereas the Parallel GCNN maintains stable predictions even at higher perturbation levels such as $\epsilon = 0.15$.

### F.2 Ablation Study

To evaluate the influence of equivariance convolution modules on the proposed model, we conducted ablation studies in default settings. The experiments compare the performance of four models—Baseline CNN, Parallel GCNN with Rotation-Equivariant Branch (Rot), Parallel GCNN with Scale-Equivariant Branch (Scale), and Parallel GCNN with Rotation- and Scale-Equivariant Branch (Rot & Scale)—on CIFAR-100 under FGSM and PGD attacks across varying perturbation levels. The Parallel GCNN with Rotation- and Scale-Equivariant Branch consistently outperformed other models, achieving the highest robustness across all settings. Rotation-equivariant convolutions demonstrated better performance than scale-equivariant convolutions individually, particularly under higher perturbation magnitudes. The Baseline CNN lacked adversarial robustness, showing the superiority of symmetry-aware architectural designs.

Under the more challenging PGD attacks, the Parallel GCNN with Rotation- and Scale-Equivariant Branch continued to outperform other models. At $\epsilon = 0.01$, it achieved 29.62% accuracy, compared to 24.72% for the Rotation-Equivariant GCNN, 21.34% for the Scale-Equivariant GCNN, and only 2.49% for the Baseline CNN. Even at $\epsilon = 0.04$, the combined model retained 6.04% accuracy,

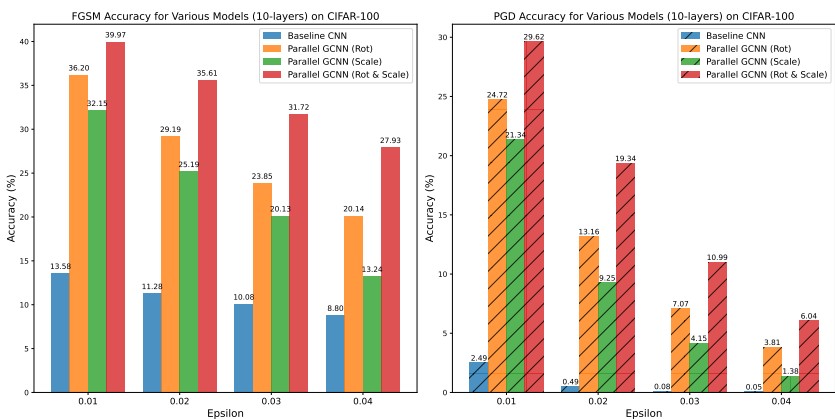

Figure 6: Ablation study results on CIFAR-100 comparing the adversarial robustness of four models under FGSM (left) and PGD (right) attacks.

while the Rotation-Equivariant GCNN and Scale-Equivariant GCNN dropped to 3.81% and 1.38%, respectively. The Baseline CNN showed minimal robustness at higher perturbations, with accuracy falling below 1%.

Our experiments reveal several key insights for the integration of the equivariance layer in the CNN model. The parallel design consistently outperforms the cascaded design in both clean and adversarial settings. By maintaining the independence of standard and equivariant properties, the parallel design achieves diverse and complementary feature representations that enhance adversarial resilience. Within the parallel design, simple concatenation of outputs consistently delivers higher robustness against FGSM and PGD attacks. This approach retains richer and more balanced feature representations, which are crucial for adversarial defense. Weighted sum fusion, while effective on clean data, is more vulnerable to adversarial attacks due to over-reliance on learned weights, which may fail to generalize under adversarial perturbations. The cascaded design exhibits diminished performance and robustness due to feature redundancy. The sequential dependence between standard and equivariant layers restricts the network's ability to capture diverse patterns, leading to reduced effectiveness in adversarial scenarios.

## G   Experiments Computing Resources

All experiments were performed on the Lawrence Supercomputer and NRP Nautilus HPC systems. Training and evaluation used a single GPU per experiment. Node specifications are provided in Table 5.

Table 5: Hardware configuration for experiments

| Component | Configuration |
| --- | --- |
| CPUs | Dual 12-core SkyLake 5000 series |
| GPUs | Nvidia Tesla P100 16GB or Nvidia RTXA6000 64GB |
| RAM | 64GB |
| SSD | 240GB |

