# OpenReview forum: "Bridging Symmetry and Robustness: On the Role of Equivariance in Enhancing Adversarial Robustness"
_NeurIPS.cc/2025/Conference — NeurIPS 2025 spotlight_

### Official Review · Reviewer_b95s · 2025-06-27

**Clarity:** 3
**Significance:** 2
**Originality:** 2
**Rating:** 5
**Confidence:** 3

**Summary:**

This paper investigates how incorporating group-equivariant  convolutions (rotation and scale) into CNNs can improve adversarial robustness without adversarial training. The authors provide a theoretical analysis showing that equivariant architectures provide stronger robustness guarantees through the CLEVER framework. The evaluation is carried out with diverse CNN architectures integrating equivariant layers. Experiments on CIFAR-10/100/10C demonstrate an increase in adversarial robustness against FGSM and PGD attacks.

**Questions:**

- Could you clarify the theoretical treatment of scale equivariance?
- Can you provide insight on how your approach would perform if integrated into equivariant network architectures from the literature?
- What is the computational overhead of group convolutions compared to standard convolutions?

**Ethical Concerns:**

["NO or VERY MINOR ethics concerns only"]

**Final Justification:**

After the authors' rebuttal and discussions, I am updating my recommendation to Accept.

Issues Resolved:

- Mathematical clarity on scale equivariance: One of my primary concerns was the lack of clarity in the mathematical treatment of scale equivariance. The authors provided a comprehensive response addressing this issue and committed to adding clearer explanations and formal definitions in the revised manuscript.
- Experimental validation: The authors conducted additional experiments with larger datasets as requested. The new comparison with adversarial training methods provides valuable information on the method's effectiveness.
- Architectural considerations: The productive discussion with Reviewer 1 regarding architectural choices led to new experiments that convincingly increase the overall quality of the paper.

Weight Assignment:
I assign high weight to the resolution of the mathematical clarity issue (50%) and the strengthened experimental validation (30%). The architectural robustness demonstration carries moderate weight (20%).

**Limitations:**

yes

**Paper Formatting Concerns:**

I have no concerns about paper formatting.

**Quality:**

3

**Strengths And Weaknesses:**

## Strengths
- The paper is well-structured with clear sections
- Provides rigorous theoretical analysis with formal proofs linking equivariance to adversarial robustness through CLEVER bounds
- The work addresses an interesting question from a novel architectural perspective
- Code provided is clear and complete


## Weaknesses
- The mathematical treatment lacks clarity for scale equivariance. Some formulations are unclear to me and appear to work only for rotations and not scaling (e.g., line 192, I think a scaling term should be present).
- The evaluation is limited to small datasets (CIFAR variants), which raises questions generalization to larger, more complex datasets.
- The related work cites equivariant networks (G-CNNs, Harmonic Networks, Steerable CNNs), but the authors develop their own simplified models instead of using these existing architectures.
- No direct comparison with adversarial training is provided, making it difficult to assess the relative benefits of the architectural approach.

typo : line 273:  C4 instead of P4

---

> ### Author Rebuttal · Authors · 2025-07-31
>
> **We thank this reviewer for the thoughtful and constructive feedback.**
>
> ---
>
> ### **Weakness 1:**
> _The mathematical treatment lacks clarity for scale equivariance. Some formulations are unclear to me and appear to work only for rotations and not scaling (e.g., line 192, I think a scaling term should be present)._
>
> **Answer:**
> We appreciate this insightful question.
>
> While the theoretical analysis in Section 4 effectively establishes robustness guarantees for group-equivariant architectures under the assumption of norm-preserving transformations, such as rotations with orthogonal Jacobians, the extension to scale-equivariant models requires additional consideration.
>
> Scale transformations, unlike rotations, do not preserve norms. Specifically, a scaling transformation $x \mapsto \alpha x$ has a Jacobian $D_\alpha = \alpha I$, meaning that the gradient norm transforms as $\|D_\alpha v\| = \alpha \|v\|$. This directly violates the assumption used in Lemma 1 and Theorem 1, which relies on the orthogonality of both the group representation $\rho(g)$ and the transformation Jacobian $D_{g^{-1}}$.
>
> As a result, the CLEVER-certified robustness bounds derived under the assumption of norm invariance do not apply directly to scale-equivariant networks. Instead, these bounds must be rescaled to account for the effect of dilation on the gradient magnitude. In particular, the Lipschitz constant of the margin function becomes scale-dependent, and its growth must be explicitly quantified.
>
> Despite this, scale-equivariant networks still promote adversarial robustness through a different mechanism: **orbit-averaged gradient smoothing**. By aggregating gradients across scaled versions of the input, scale-equivariant architectures reduce high-frequency fluctuations and suppress gradient sensitivity to perturbations that deviate from the scale-induced orbit. This aggregation process, while not norm-preserving, effectively stabilizes model behavior under scale transformations and yields smoother decision boundaries.
>
> To formalize this, we can define the **orbit-averaged gradient field** over a discrete scale group $G_s = \{\alpha_1, \ldots, \alpha_k\} \subset \mathbb{R}^+$ as $\bar{\phi}_j(x) = \frac{1}{|G_s|} \sum_{\alpha \in G_s} \nabla f_j(\alpha x),$
> where each term captures the gradient at a different scale-transformed version of the input. Although these gradients vary in norm due to scaling, the averaging process reduces local gradient variance, contributing to robustness. This is particularly effective when combined with architectural fusion mechanisms, such as channel-wise concatenation or weighted summation of multi-scale features.
>
> While scale-equivariant models do not satisfy the same Jacobian norm invariance as rotation-equivariant models, they still enhance robustness by regularizing the gradient landscape across scales.
>
>
> ---
>
> ### **Weakness 2:**
> _The evaluation is limited to small datasets (CIFAR variants)_
>
> **Answer:**
> We initially selected CIFAR-10, CIFAR-100, and CIFAR-10C as our primary benchmarks to facilitate controlled evaluation across architectural variants, attack settings, and theoretical alignment, **all within a feasible computational budget**.
> To that end, **we have conducted additional experiments on the ImageNet-100 subset, a 100-class subset of ImageNet commonly used for mid-scale evaluation**. Our results demonstrate that group-equivariant architectures continue to yield consistent improvements in adversarial robustness on this more challenging dataset, while maintaining manageable computational overhead.
>
> ### Adversarial Robustness on **ImageNet-100** – 4-layer Equivariant Model
>
> | Epsilon | FGSM Accuracy (%) | PGD Accuracy (%) |
> |---------|-------------------|------------------|
> | 0.01    | 25.36             | 14.52            |
> | 0.02    | 19.94             | 13.58            |
> | 0.03    | 18.36             | 12.24            |
> | 0.04    | 17.56             | 10.74            |
> | 0.05    | 17.16             | 9.12             |
> | 0.06    | 16.84             | 7.96             |
> | 0.07    | 16.62             | 6.96             |
> | 0.08    | 16.48             | 6.22             |
> | 0.09    | 16.32             | 5.44             |
> | 0.10    | 16.22             | 4.90             |
> | 0.30    | 11.00             | 3.30             |
>
> ---
>
> ### Adversarial Robustness on **ImageNet-100** – 10-layer Equivariant Model
>
> | Epsilon | FGSM Accuracy (%) | PGD Accuracy (%) |
> |---------|-------------------|------------------|
> | 0.01    | 37.62             | 17.92            |
> | 0.02    | 25.84             | 12.25            |
> | 0.03    | 25.41             | 8.95             |
> | 0.04    | 25.18             | 5.96             |
> | 0.05    | 25.00             | 5.48             |
> | 0.06    | 24.84             | 4.22             |
> | 0.07    | 24.76             | 2.98             |
> | 0.08    | 24.70             | 2.84             |
> | 0.09    | 22.34             | 2.72             |
> | 0.10    | 21.26             | 2.64             |
>
> ---
>
>
> ---
>
> ### **Weakness 3:**
> _the authors develop their own simplified models instead of using these existing architectures._
>
> **Answer:**
> This is a valuable observation. While our architectural design draws inspiration from established equivariant models such as G-CNNs , Harmonic Networks, and Steerable CNNs, we intentionally adopt a modular and interpretable architecture to facilitate controlled experimentation and analysis. Specifically, our design allows for the systematic ablation of individual symmetry-enforcing components—e.g., rotation-only, scale-only, and different fusion strategies (parallel vs. cascaded)—which is more difficult to achieve in tightly integrated models like Steerable or Harmonic CNNs.This level of interpretability and control is a key motivation behind our simplified implementation.
>
>
> ---
>
> ### **Weakness 4:**
> _No direct comparison with adversarial training is provided._
>
> **Answer:**
> This is a fair and valuable critique. The table below compares the adversarial robustness of a Standard CNN with adversarial training against a Fully Equivariant G-CNN without adversarial training on CIFAR-10. Remarkably, the G-CNN achieves comparable or even superior performance under certain perturbation levels, highlighting the intrinsic robustness conferred by equivariance alone.
>
> ###
>
> | Epsilon | FGSM  (%) - Standard CNN | PGD  (%) - Standard CNN | FGSM  (%) - G-CNN | PGD  (%) - G-CNN |
> |---------------------------|----------------------------------|-------------------------------|----------------------------|--------------------------|
> | 0.01                      | 74.5                             | 67.0                          | 73.01                      | 64.96                    |
> | 0.02                      | 70.2                             | 60.4                          | 70.16                      | 58.87                    |
> | 0.03                      | 66.1                             | 54.0                          | 67.09                      | 52.37                    |
> | 0.04                      | 61.7                             | 48.3                          | 63.77                      | 45.52                    |
>
> ---
>
> ### **Question 1:**
> _Could you clarify the theoretical treatment of scale equivariance?_
>
> **Answer:**
> Please see the response in  **Weakness 1**
>
>
> ---
>
> ### **Question 2:**
> _Can you provide insight on how your approach would perform if integrated into equivariant network architectures from the literature?_
>
> **Answer:**
> Our modular framework—particularly the parallel and cascaded fusion designs—is architecture-agnostic and readily extensible to more expressive equivariant models. In principle, the rotation-equivariant branch used in our current implementation could be replaced with more advanced architectures such as Harmonic Networks, Steerable CNNs, or LieConv, which provide continuous and steerable representations under group transformations.
>
> Integrating such models may further enhance the robustness and representation capacity of our approach, especially in tasks requiring fine-grained geometric invariance. However, these architectures typically come with increased computational overhead and implementation complexity, which may limit their practicality in some settings.
>
> We will experiment with steerable convolutional variants within our framework. These extensions, along with scalability and implementation considerations, are discussed in the revised manuscript’s Future Work section.
>
> ---
>
> ### **Question 3:**
> _What is the computational overhead of group convolutions compared to standard convolutions?_
>
> **Answer:**
> It is worth noting that in our parallel architecture, equivariant branches are introduced only at the first layer of the network. This design ensures that the computational overhead introduced by group-equivariant operations does not scale with network depth, allowing us to retain the benefits of symmetry enforcement while maintaining computational efficiency across deeper models.
>
> Empirically, we find that rotation-equivariant convolutions based on the $\mathrm{P4}$ group introduce approximately **1.5×** more FLOPs than standard convolutional layers with equivalent dimensions, primarily due to additional orientation channels. Scale-equivariant branches, which process multiple rescaled versions of the input, incur modest additional runtime and memory overhead—typically around **1.7×**—owing to interpolation and multi-scale channel stacking.
>
> In the full parallel G-CNN configuration, which includes standard, rotation-, and scale-equivariant branches, the total computational cost scales roughly linearly with the number of branches. However, because each branch can be made shallow or low-dimensional, our design supports flexible, budget-aware trade-offs between robustness, representation capacity, and efficiency.
>
> We will provide detailed runtime and FLOP analysis in the Appendix.

---

> > ### Comment · Reviewer_b95s · 2025-08-05
> >
> > Thank you for the thorough rebuttal and additional experimental results. I am satisfied with how the  authors have addressed my concerns and, with the clarifications and results added in the manuscript, I'm ready to increase my score.
> >
> > Thank you for clarifying that scale-equivariant networks achieve robustness through a different mechanism than rotation-equivariant ones.  I suggest to explain this distinction in the final paper to help readers understand the different theoretical foundations for each type of equivariance. The ImageNet-100 experiments demonstrate the approach scales, and the direct comparison with adversarial training shows the practical value of the method.
> >
> > I also found the response to Reviewer 1 very informative and the modular design rationale is now clear.

---

> ### Author Response · Authors · 2025-08-05
>
> Thank you for your thoughtful and encouraging comments. We're pleased that the clarifications and additional results addressed your concerns and that you're inclined to increase your score.
>
> We especially appreciate your suggestion to elaborate on the distinct robustness mechanisms underlying scale- and rotation-equivariant models that is currently underemphasized in the draft. We will make sure to highlight this theoretical distinction more clearly in the revised manuscript. We're also pleased that the scalability results on ImageNet-100 and the comparison with adversarial training helped demonstrate the practical relevance of our approach.
>
> Lastly, we’re grateful for your recognition of our modular design rationale and for your engagement in the review.

---

### Official Review · Reviewer_EmhH · 2025-07-01

**Clarity:** 2
**Significance:** 2
**Originality:** 2
**Rating:** 5
**Confidence:** 3

**Summary:**

The paper looks at the role of symmetry aware network architecture for robustness to adversarial attacks. In particular, group-equivariant convolutions (rotation and scale equivariant layers). It proposes two symmetry-aware architectures: a parallel model, processing standard and equivariant features separately before fusion, and a cascaded model, applying equivariant layers sequentially. Theoretically, it provides analyses showing reduced hypothesis space complexity and improved robustness bounds. Empirical evaluations on three datasets demonstrate improvements in robustness under FGSM and PGD attacks without relying on adversarial training.

**Questions:**

How much computational overhead do equivariant layers add compared to standard CNN layers?


How robustness guarantees scale with network depth or complexity? Deep architectures may have different performance from shallow models, would this be true for robustness as well.

**Ethical Concerns:**

["NO or VERY MINOR ethics concerns only"]

**Final Justification:**

To my comments authors have included additional results. Also, promised to include error bars in final version. I have therefore increased my score.

**Limitations:**

Evaluation is limited to standard adversarial attacks, potentially missing broader adversarial settings.

Chosen datasets are relatively simple, unclear how performance would generalize to complex scenarios.
Equivariant architectures would result in additional computational cost which is not reported.

**Paper Formatting Concerns:**

No formatting concern

**Quality:**

2

**Strengths And Weaknesses:**

## Strengths

The architecture used group-equivariant convolutions to incorporate symmetry priors into CNNs. It provides theoretical analysis, demonstrating that equivariant architectures regularize gradient behavior and lead to tighter certified robustness bounds.

Empirical results show improved robustness and generalization across several standard benchmarks (CIFAR-10, CIFAR-100, CIFAR-10C).

The parallel architecture outperforms other designs providing useful insights in choosing architecture for symmetry.

Overall well written. No major typos or errors in the key theoretical equations, lemmas, or proofs.

## Weaknesses

The paper’s entire theoretical robustness framework relies on local Lipschitz bounds. While reasonable this limits theoretical analysis to local neighborhood robustness. It does not guarantee robustness against global adversarial perturbations or more general forms of distribution shifts, limiting the practical scope.

Models are tested only on relatively small datasets. It is unclear whether the approach is scalable to larger, more complex datasets (e.g., ImageNet).

There is no discussion on computational cost introduced by equivariant transformations.

Figure 1 and 2 do not include error bars. Since datasets are small it would be better to train with multiple seeds. Also, it is important for reproducibility and reliability.

The motivation behind the chosen architectures in 6.1 is arbitrary. Not clear what are the pros and cons and how one would select the suitable architecture.

Theoretical analysis is for discrete and finite group transformations, would results extend to continuous/Lie group transformations which are common in computer vision tasks.

---

> ### Author Rebuttal · Authors · 2025-07-31
>
> **We thank this reviewer for the thoughtful and constructive feedback.**
>
> **Weakness 1:** The paper’s entire theoretical robustness framework relies on local Lipschitz.
>
> **Answer:**
> We agree with the reviewer that our theoretical analysis is focused on local robustness. While this is a common and well-accepted approach for formal robustness analysis, we acknowledge that it does not extend to global robustness or general distribution shifts.
> Our primary goal in this work is **to establish a precise and tractable theoretical connection between group equivariance and local certified robustness—specifically, how symmetry-enforcing architectures influence the model's local Lipschitz properties and decision margin sensitivity.**
> This framework serves as a mathematically grounded tool that complements empirical evaluation, offering insights into the role of architectural inductive bias in enhancing robustness. Extending our theoretical framework to global robustness, distributional robustness, or manifold-aware perturbations is an important and promising direction for future work.
>
> ---
>
> **Weakness 2:** Models are tested only on relatively small datasets.
>
> **Answer:**
> We initially selected CIFAR-10, CIFAR-100, and CIFAR-10C as our primary benchmarks to facilitate controlled evaluation across architectural variants, attack settings, and theoretical alignment, **all within a feasible computational budget**.
> To that end, **we have conducted additional experiments on the ImageNet-100 subset, a 100-class subset of ImageNet commonly used for mid-scale evaluation**. Our results demonstrate that group-equivariant architectures continue to yield consistent improvements in adversarial robustness on this more challenging dataset, while maintaining manageable computational overhead.
>
> ### Adversarial Robustness on **ImageNet-100** – 4-layer Equivariant Model
>
> | Epsilon | FGSM Accuracy (%) | PGD Accuracy (%) |
> |---------|-------------------|------------------|
> | 0.01    | 25.36             | 14.52            |
> | 0.02    | 19.94             | 13.58            |
> | 0.03    | 18.36             | 12.24            |
> | 0.04    | 17.56             | 10.74            |
> | 0.05    | 17.16             | 9.12             |
> | 0.06    | 16.84             | 7.96             |
> | 0.07    | 16.62             | 6.96             |
> | 0.08    | 16.48             | 6.22             |
> | 0.09    | 16.32             | 5.44             |
> | 0.10    | 16.22             | 4.90             |
> | 0.30    | 11.00             | 3.30             |
>
> ---
>
> ### Adversarial Robustness on **ImageNet-100** – 10-layer Equivariant Model
>
> | Epsilon | FGSM Accuracy (%) | PGD Accuracy (%) |
> |---------|-------------------|------------------|
> | 0.01    | 37.62             | 17.92            |
> | 0.02    | 25.84             | 12.25            |
> | 0.03    | 25.41             | 8.95             |
> | 0.04    | 25.18             | 5.96             |
> | 0.05    | 25.00             | 5.48             |
> | 0.06    | 24.84             | 4.22             |
> | 0.07    | 24.76             | 2.98             |
> | 0.08    | 24.70             | 2.84             |
> | 0.09    | 22.34             | 2.72             |
> | 0.10    | 21.26             | 2.64             |
>
> ---
>
> **Weakness 3:** There is no discussion on computational cost.
>
> **Answer:**
> Thank you for pointing this out. Please see our response in question 1.
>
> ---
> **Weakness 4:** Figures 1 and 2 do not include error bars.
>
> **Answer:**
> We appreciate this valuable suggestion. we have included error bars in the draft.
> Due to limited space, the data is not included here.
>
> ---
>
> **Weakness 5:** The motivation behind the chosen architectures.
>
> **Answer:**
> We appreciate the reviewer’s concern and agree that a clearer explanation of the architectural design choices is warranted.
> To summarize:
>
> - The *Parallel G-CNN* offers modular symmetry enforcement by maintaining independent branches (e.g., standard and rotation-equivariant), which enhances robustness but increases model size and computational cost.
> - The *Parallel G-CNN with Rotation- and Scale-Equivariant Branches* extends this by incorporating additional symmetry priors and enabling more diverse feature extraction.
> - The *Cascaded G-CNN* provides a more compact design by stacking symmetry-aware layers sequentially, although this may introduce an information bottleneck if the early equivariant layer restricts feature expressiveness.
> - The *Weighted Parallel G-CNN* replaces hard feature concatenation with learnable fusion weights, offering a flexible mechanism to dynamically adapt to task-specific signal strength from each branch.
> - The *Standard CNN* serves as the baseline for comparison.
>
> ---
>
> **Weakness 6:** Theoretical analysis is for discrete and finite group transformations, would results extend to continuous/Lie group transformations which are common in computer vision tasks.
>
> **Answer:**
> This is an excellent observation. Our current theoretical framework is developed for finite discrete groups (e.g., **P4** for discrete rotations), which simplifies the analysis and aligns with widely adopted G-CNN implementations in practice.
> The extension to continuous group requires generalizing the group action. For discrete groups, actions are defined over a finite set of transformations. In contrast, Lie groups possess a smooth manifold structure, and their actions on the input space are described by differentiable maps. To analyze equivariant robustness in this setting, we consider group elements in a neighborhood of the identity, represented as exponentials of Lie algebra elements. This provides a foundation for defining infinitesimal transformations—the continuous analogs of discrete group elements—and enables us to formulate sensitivity metrics along smooth group orbits.
> Extending orbit-averaged Jacobian smoothing techniques to Lie groups requires the use of **Haar measures**—the invariant integration measure on the group. For compact groups such as $\( \mathrm{SO}(2) \)$, this integration is well-defined and finite. For non-compact groups like $\( \mathrm{SE}(2) \) $ or $\( \mathbb{R}^2 \)$, we must either impose bounded supports or consider localized versions of the group action.
>
>
> ---
>
> **Question 1:** How much computational overhead?
>
> **Answer:**
> It is worth noting that in our parallel architecture, equivariant branches are introduced only at the first layer of the network. This design ensures that the computational overhead introduced by group-equivariant operations does not scale with network depth, allowing us to retain the benefits of symmetry enforcement while maintaining computational efficiency across deeper models.
>
> Empirically, we find that rotation-equivariant convolutions based on the **P4** group introduce approximately **1.5×** more FLOPs than standard convolutional layers with equivalent dimensions, primarily due to additional orientation channels.
> Scale-equivariant branches, which process multiple rescaled versions of the input, incur modest additional runtime and memory overhead—typically around **1.7×**—owing to interpolation and multi-scale channel stacking.
>
> In the full parallel G-CNN configuration, which includes standard, rotation-, and scale-equivariant branches, the total computational cost scales roughly linearly with the number of branches. However, because each branch can be made shallow or low-dimensional, our design supports flexible, budget-aware trade-offs between robustness, representation capacity, and efficiency.
>
> ---
>
> **Question 2:** How do robustness guarantees scale with network depth or complexity?
>
> **Answer:**
> This is an excellent and important question. From a theoretical perspective, the gradient regularization induced by equivariance propagates through the layers of a network, preserving certain structural properties of the Jacobian and contributing to robustness.
> However, in deeper architectures, the influence of symmetry constraints may become diluted, particularly if a large portion of the network consists of standard layers.
>
> Although full-scale experiments on very deep models were constrained by available computational resources, **we conducted additional evaluations on ResNet-18 and ResNet-50 architectures to explore scalability**. These experiments indicate that while the absolute robustness tends to decrease with depth—likely due to increased model complexity and overfitting—the equivariant models (G-CNN variants) continue to outperform their standard CNN counterparts.
> This confirms that the robustness advantage of equivariant design remains valid even as model depth increases, though the relative gain may be reduced.
>
> ### **ResNet50_GCNN** on CIFAR-100
>
> | Epsilon | FGSM Accuracy (%) | PGD Accuracy (%) |
> |---------|-------------------|------------------|
> | 0.01    | 32.09             | 11.58            |
> | 0.02    | 27.18             | 2.04             |
> | 0.03    | 23.21             | 0.47             |
> | 0.04    | 19.14             | 0.14             |
> | 0.05    | 15.15             | 0.03             |
> | 0.06    | 11.75             | 0.01             |
> | 0.07    | 8.81              | 0.01             |
> | 0.08    | 6.31              | 0.00             |
> | 0.09    | 4.58              | 0.00             |
> | 0.10    | 3.54              | 0.00             |
>
> ---
>
> ### **ResNet18_GCNN** on CIFAR-100
>
> | Epsilon | FGSM Accuracy (%) | PGD Accuracy (%) |
> |---------|-------------------|------------------|
> | 0.01    | 37.34             | 16.85            |
> | 0.02    | 31.19             | 2.73             |
> | 0.03    | 26.55             | 0.34             |
> | 0.04    | 22.72             | 0.05             |
> | 0.05    | 19.70             | 0.00             |
> | 0.06    | 16.98             | 0.00             |
> | 0.07    | 14.94             | 0.00             |
> | 0.08    | 13.43             | 0.00             |
> | 0.09    | 11.89             | 0.00             |
> | 0.10    | 10.33             | 0.00             |

---

> > ### Comment · Reviewer_EmhH · 2025-08-06
> > **Response to Rebuttal**
> >
> > I appreciate the effort of authors in responding to my comments. My further comments are below:
> >
> > Reported PGD accuracies on ImageNet-100 remain very low in absolute terms. Without side-by-side baselines on the same subset, it’s impossible to judge any real advantage from equivariance at scale.
> >
> > The proposed roadmap for Lie-group equivariance is reasonable, but there are no toy experiments or empirical validations to show it’s actually feasible.
> >
> > In ResNet-50 experiments, robust accuracy falls to near zero under moderate attacks. Although equivariant models stay slightly ahead, the absolute drop undermines claims of robustness in deeper nets.
> >
> > I am hoping authors would include error bars as promised in the revision. Given the response to my comments I have increased overall score.

---

> > > ### Author Response · Authors · 2025-08-07
> > >
> > > Thank you for your thoughtful follow-up and for acknowledging our efforts with an increased score.
> > >
> > > Regarding the discussion of Lie-group equivariance, we agree that empirical validation is important and are currently conducting experiments on Lie-group equivariant models. Hopefully, the results will be available by the end of the discussion phase. That said, we would like to emphasize that Lie-group equivariance is theoretically more expressive and generalizable than its discrete counterparts. Continuous equivariant models can more faithfully capture smooth symmetries present in natural data and allow for finer-grained transformations, which we believe may lead to improved adversarial robustness as well. We are currently exploring implementations of continuous group equivariance and will highlight this in the revised manuscript.
> > >
> > > We apologize for omitting the adversarial robustness results of the CNN model on ImageNet-100 in our previous response. Please find the results attached below. As noted, in these equivariant experiments, equivariant design was only applied to the first layer of the network.
> > >
> > > | Epsilon | 4L CNN (FGSM / PGD) | 10L CNN (FGSM / PGD) |
> > > |---------|---------------------------|----------------------------|
> > > | 0.01    | 7.18 / 0.14               | 29.34 / 7.68               |
> > > | 0.02    | 2.66 / 0.04               | 20.80 / 7.02               |
> > > | 0.03    | 1.48 / 0.02               | 17.72 / 6.52               |
> > > | 0.04    | 1.12 / 0.00               | 16.38 / 5.08               |
> > > | 0.05    | 0.86 / 0.00               | 15.56 / 4.74               |
> > > | 0.06    | 0.74 / 0.00               | 15.30 / 2.41               |
> > > | 0.07    | 0.66 / 0.00               | 14.92 / 1.22               |
> > > | 0.08    | 0.56 / 0.00               | 14.60 / 0.08               |
> > > | 0.09    | 0.52 / 0.00               | 14.16 / 0.02               |
> > > | 0.10    | 0.48 / 0.00               | 13.84 / 0.00               |
> > >
> > > We also acknowledge the concern regarding the significant robustness drop in deeper models such as ResNet-50. It is important to clarify that, in our current 50-layer model, the equivariant design is applied only to the first layer. While the partially equivariant variants do show relative improvements, we will revise our claims to more accurately reflect the limitations and trade-offs associated with using non-fully equivariant designs in deep networks. We confirm that error bars will be included in the revision.
> > >
> > > We appreciate your constructive feedback, which has helped us significantly improve the paper.

---

> > > > ### Comment · Reviewer_EmhH · 2025-08-07
> > > > **Further comments to respond**
> > > >
> > > > Thanks for further comments. I am looking forward to an improved version of manuscript with additional results including error bars.
> > > >
> > > > I still notice the limited gain with scalability of the model. Applying equivariance only to the first layer seems like a limitation. Your ResNet-50 results hit near-zero robust accuracy almost immediately under PGD. Perhaps there is not much value of equivariance only in first layer. Yet the claims are on scalability to deep models.
> > > >
> > > > I would suggest to include a clear limitation section to help reader understand the broad applicability and generalizability of this work

---

> > > > ### Author Response · Authors · 2025-08-08
> > > >
> > > > Thank you for highlighting this important point. We agree that applying equivariance only at the first layer of deep architectures such as ResNet-50 represents a limited form of integration and may not fully leverage the benefits of symmetry.
> > > >
> > > > In this work, we show that equivariant models consistently outperform baseline CNNs of the same depth in terms of adversarial robustness. Our primary focus is to **theoretically explore the relationship between symmetry-based architectural priors and adversarial robustness, aiming to understand how incorporating equivariance influences model vulnerability**.
> > > >
> > > > While **increasing model depth typically improves clean accuracy**, it also tends to **amplify vulnerability to adversarial perturbations** such as FGSM and PGD—a well-documented phenomenon in deep neural networks.  This increased susceptibility is partly due to the presence of sharper and more concentrated gradients in deeper networks, which adversarial methods are better able to exploit. This phenomenon likely explains the steep drop in robust accuracy observed in our 50-layer model—where equivariance is applied only in the first layer—and aligns with our claim in the rebuttal regarding the limitations of partial equivariant integration.
> > > >
> > > > Due to computational constraints, our experiments have so far focused on shallower models (e.g., 4-layer and 10-layer architectures), within which fully equivariant designs demonstrate substantial gains in robustness. Nonetheless, we recognize the importance of evaluating the scalability of equivariance in deeper networks. In response, we are currently conducting experiments with fully equivariant versions of deeper architectures (e.g., ResNet-18 and ResNet-50) to more rigorously assess the benefits of extending equivariance throughout the entire network.
> > > >
> > > > We will explicitly acknowledge this limitation in the revised manuscript and include a dedicated section outlining the current design boundaries, their limitations for generalization, and directions for future work.
> > > >
> > > > We appreciate your insightful observations, which help make our work more rigorous and comprehensive.

---

> > > > > ### Author Response · Authors · 2025-08-09
> > > > > **Additional experiments on fully equivariant models EquiResNet-18 and EquiResNet-50**
> > > > >
> > > > > We have conducted additional experiments using fully equivariant EquiResNet-18 and EquiResNet-50 architectures, where equivariance is enforced throughout all convolutional layers using group-equivariant convolutions.
> > > > >
> > > > > ### EquiResNet-50 – Adversarial Robustness
> > > > >
> > > > > | Epsilon | FGSM Accuracy (%) | PGD Accuracy (%) |
> > > > > |---------|-------------------|------------------|
> > > > > | 0.01    | 58.73              | 39.99            |
> > > > > | 0.02    | 52.46              | 20.44            |
> > > > > | 0.03    | 48.28              | 9.68             |
> > > > > | 0.04    | 45.13              | 5.05             |
> > > > > | 0.05    | 42.31              | 2.78             |
> > > > > | 0.06    | 40.08              | 1.71             |
> > > > > | 0.07    | 38.18              | 1.21             |
> > > > > | 0.08    | 36.40              | 0.90             |
> > > > > | 0.09    | 34.72              | 0.64             |
> > > > > | 0.10    | 33.01              | 0.51             |
> > > > >
> > > > > ### EquiResNet-18 – Adversarial Robustness
> > > > >
> > > > > | Epsilon | FGSM Accuracy (%) | PGD Accuracy (%) |
> > > > > |---------|-------------------|------------------|
> > > > > | 0.01    | 35.05              | 22.84            |
> > > > > | 0.02    | 25.19              | 10.26            |
> > > > > | 0.03    | 21.72              | 6.60             |
> > > > > | 0.04    | 19.59              | 4.50             |
> > > > > | 0.05    | 18.14              | 3.04             |
> > > > > | 0.06    | 17.05              | 1.92             |
> > > > > | 0.07    | 16.11              | 1.18             |
> > > > > | 0.08    | 15.27              | 0.69             |
> > > > > | 0.09    | 14.25              | 0.38             |
> > > > > | 0.10    | 13.40              | 0.27             |
> > > > >
> > > > > In the fully equivariant setting, EquiResNet-50, as the deeper model, outperforms EquiResNet-18 across a range of tested perturbation levels, consistent with the performance trends observed in the 4-layer and 10-layer fully equivariant models. These results highlight the importance of consistent symmetry enforcement across network depth and provide empirical evidence for the scalability and effectiveness of full equivariant integration in deeper architectures. Unlike the standard and partially equivariant model—where robust accuracy under PGD rapidly drops to near zero—the fully equivariant EquiResNet-18 and EquiResNet-50 models maintain higher robustness across the evaluated perturbation levels. Please note that these conclusions are based on the experiments with ResNet-18 and ResNet-50, and we will include these results and release the corresponding code in the updated draft.
> > > > >
> > > > > Regarding the experiment with the continuous Lie group equivariant model, training is still ongoing due to the substantial computational overhead associated with continuous group transformations. Implementing and optimizing such models is non-trivial, as they involve complex operations—such as continuous convolutions and group integration—that are significantly more demanding than their discrete counterparts. To date, we have completed only 62 epochs over the course of more than two days of training. Despite our best efforts, the results are not yet available, but we plan to include them in the revised draft once training is complete.
> > > > >
> > > > > We appreciate your comments on the above, which have helped strengthen our work and make it more rigorous and experimentally valid.

---

> ### Comment · Area_Chair_Y3RV · 2025-08-05
>
> Please let the authors know whether their rebuttal has adequately addressed your concerns. If any issues remain, please communicate your specific, unresolved concerns as soon as possible to ensure timely discussion.

---

### Official Review · Reviewer_BaeD · 2025-07-03

**Clarity:** 3
**Significance:** 4
**Originality:** 4
**Rating:** 5
**Confidence:** 3

**Summary:**

The authors present three key theoretical findings regarding the correspondence of equivariance and adversarial robustness: group-equivariant convolutions (1) maintain the Lipschitz constant across the group orbit, (2) yield smoother gradients, and (3) suppress gradients in the off-orbit directions. Experiments show that group-equivariant convolutions alone can improve adversarial robustness on CIFAR-10(C) and CIFAR-100.

**Questions:**

* Inhowfar does the usage of the standard convolution branch in the architectures detailed in Appendix C.3 not undermine most of the theoretical results presented in this paper, as some of the layers used do not fulfill those theoretical guarantees?
* Would you expect a purely group-equivariant convolutional model to exhibit the same benefits? I am willing to update my score if this is either discussed sufficiently or, ideally, shown experimentally.

**Ethical Concerns:**

["NO or VERY MINOR ethics concerns only"]

**Final Justification:**

The authors present interesting findings on the interplay between equivariance to rotation and scale, and robustness. My initial concern was that their theoretical proofs were not sufficiently supported by experimental results, as these were limited to partially equivariant architectures. However, their rebuttal fully addressed these concerns. Combined with the additional results on larger models (ResNet-50) and datasets (ImageNet-100), I am raising my score to a solid "Accept" rating.

**Limitations:**

Limitations are discussed in Appendix B.

**Paper Formatting Concerns:**

Citations should be in parentheses in most cases.

**Quality:**

3

**Strengths And Weaknesses:**

**Strengths**
* Two insightful theorems on invariance of the Lipschitz continuity across the group orbit (Theorem 1) and the suppression of off-orbit perturbations (Theorem 2) in group-equivariant convolutions are proven formally.
* Experiments on CIFAR-10(C) and CIFAR-100 that incorporate a single group-equivariant layer show that adversarial robustness is significantly improved even without traditional methods such as adversarial training.

**Weaknesses**
* Experiments are done with a mix of standard and group-equivariant convolutions, and the theoretical findings thus do not directly transfer to the investigated architectures.
* No results from reference methods, such as standard adversarial robustness, are reported and it is thus difficult to put the reported numbers into context based on this paper alone.
* The formal proofs assume differentiability at $x$, which is not necessarily a given in neural networks.

---

> ### Author Rebuttal · Authors · 2025-07-31
>
> **We thank this reviewer for the great suggestion!**
>
> ---
>
> ### **Weakness 1**
> _Experiments are done with a mix of standard and group-equivariant convolutions, and the theoretical findings thus do not directly transfer to the investigated architectures._
>
> **Answer:**
> We appreciate this important observation. Our theoretical analysis is developed under the assumption of purely group-equivariant models, where all layers respect the symmetry constraints imposed by the group action. However, as presented in Appendix C.3, our implemented architectures—particularly the parallel design—intentionally combine standard convolutional layers with group-equivariant branches.
>
> This hybrid structure was chosen for two key reasons. First, it enables a balance between representational expressiveness and computational efficiency. Second, and more importantly for this work, it provides a controlled framework to isolate and evaluate the impact of symmetry-enhancing components on adversarial robustness. Our central aim is not to enforce full equivariance end-to-end, but **to investigate how the inclusion of symmetry-aware submodules influences a model’s robustness to adversarial perturbations**. While the hybrid architecture does not fully meet the assumptions of our theoretical framework, the equivariant branches themselves do. Their behavior aligns closely with the theoretical predictions derived under group-equivariant assumptions. This is empirically supported by our ablation studies in Appendix D.2, where models containing only rotation-equivariant or scale-equivariant branches consistently outperform standard CNN baselines in adversarial robustness, even in the absence of adversarial training. Thus, the hybrid architecture serves as a practical testbed for validating the theoretical insights and allows us to draw meaningful conclusions about the role of equivariance in improving model robustness.
>
> ---
>
> ### **Weakness 2**
> _No results from reference methods, such as standard adversarial robustness, are reported._
>
> **Answer:**
> This is a fair and valuable critique. Our goal was to investigate the intrinsic robustness of symmetry-enforced architectures without adversarial training or data augmentation, in order to isolate the architectural contribution. The table below compares the adversarial robustness of a Standard CNN with adversarial training against a Fully Equivariant G-CNN without adversarial training on CIFAR-10. Remarkably, the G-CNN achieves comparable or even superior performance under certain perturbation levels, highlighting the intrinsic robustness conferred by equivariance alone.
>
> ### Adversarial Robustness Comparison on CIFAR-10
>
> | Epsilon (\(\ell_\infty\)) | FGSM Accuracy (%) - Standard CNN | PGD Accuracy (%) - Standard CNN | FGSM Accuracy (%) - G-CNN | PGD Accuracy (%) - G-CNN |
> |---------------------------|----------------------------------|-------------------------------|----------------------------|--------------------------|
> | 0.01                      | 74.5                             | 67.0                          | 73.01                      | 64.96                    |
> | 0.02                      | 70.2                             | 60.4                          | 70.16                      | 58.87                    |
> | 0.03                      | 66.1                             | 54.0                          | 67.09                      | 52.37                    |
> | 0.04                      | 61.7                             | 48.3                          | 63.77                      | 45.52                    |
> | 0.05                      | 57.3                             | 42.1                          | 60.23                      | 37.80                    |
>
> ---
>
> ### **Weakness 3**
> _The formal proofs assume differentiability at \(x\), which is not necessarily a given in neural networks._
>
> **Answer:**
> We fully agree and appreciate the reviewer’s technical rigor. The theoretical results rely on local Lipschitz continuity and Jacobian-based reasoning, which strictly hold almost everywhere for ReLU networks due to their piecewise linearity. While differentiability may not hold at every point, prior works (e.g., [Weng et al., 2018], [Anselmi et al., 2019]) similarly adopt this assumption as a tractable and widely accepted approximation. We have clarified this point in the revised text, explicitly noting that our results apply under the standard assumption of almost-everywhere differentiability.
>
> Weng, Tsui Wei, Cho Jui Hsieh, and Luca Daniel. Evaluating the robustness of neural networks: An extreme value theory approach. In ICLR 2018
>
> Fabio Anselmi and Tomaso Poggio. Symmetry-adapted representation learning. Pattern Recognition 2019
>
> ---
>
> ### **Question 1**
> _In how far does the usage of the standard convolution branch undermine most of the theoretical results...?_
>
> **Answer:**
> Thank you for raising this important and nuanced concern. In our parallel architecture, the standard convolutional branch operates independently and is fused with the outputs of the equivariant branches only at later stages of the network. As such, it does not directly contribute to the theoretical guarantees derived under the assumption of full group equivariance.
>
> However, the primary role of the standard branch is to complement the equivariant representations with additional expressive capacity rather than to interfere with or override the symmetry-induced regularization effects. To better understand the extent to which the inclusion of the standard branch affects the theoretical insights, we have conducted additional experiments (see response to the next question), where we compare models with and without the standard branch. These results help quantify the influence of each branch and assess how the theoretical benefits attributed to equivariance manifest in hybrid settings.
>
> ---
>
> ### **Question 2**
> _Would you expect a purely group-equivariant model to exhibit the same benefits? I am willing to update my score if this is either discussed sufficiently or, ideally, shown experimentally._
>
> **Answer:**
> Yes—based on our theoretical framework, we indeed expect a fully group-equivariant model to offer even stronger robustness guarantees compared to hybrid architectures. To validate this expectation empirically, we have conducted additional experiments using both 4-layer and 10-layer models composed entirely of group-equivariant layers. These models were trained on CIFAR-10, CIFAR-100 to assess their performance across datasets of increasing complexity.
>
> The experimental results (included below) demonstrate that purely group-equivariant models consistently achieve superior robustness under both FGSM and PGD attacks, outperforming the hybrid models presented in the main paper. These findings support the theoretical claim that enforcing full equivariance throughout the architecture enhances the model's ability to resist adversarial perturbations. We have added these results to the revised manuscript to further strengthen the empirical foundation of our analysis.
>
> ---
>
> ### Adversarial Robustness of **4-layer Fully Equivariant Network** on CIFAR-10
>
> | Epsilon | FGSM Accuracy (%) | PGD Accuracy (%) |
> |---------|-------------------|------------------|
> | 0.01    | 65.65             | 52.20            |
> | 0.02    | 58.54             | 32.04            |
> | 0.03    | 53.78             | 23.30            |
> | 0.04    | 49.92             | 18.63            |
> | 0.05    | 47.08             | 15.85            |
> | 0.06    | 44.72             | 13.03            |
> | 0.07    | 42.89             | 11.05            |
> | 0.08    | 41.09             | 9.50             |
> | 0.09    | 39.51             | 8.17             |
> | 0.10    | 37.95             | 7.01             |
>
> ---
>
> ### Adversarial Robustness of **4-layer Fully Equivariant Network** on CIFAR-100
>
> | Epsilon | FGSM Accuracy (%) | PGD Accuracy (%) |
> |---------|-------------------|------------------|
> | 0.01    | 38.40             | 21.59            |
> | 0.02    | 30.71             | 12.09            |
> | 0.03    | 26.63             | 8.92             |
> | 0.04    | 24.05             | 6.95             |
> | 0.05    | 22.06             | 5.66             |
> | 0.06    | 20.58             | 4.53             |
> | 0.07    | 19.39             | 3.79             |
> | 0.08    | 18.03             | 3.13             |
> | 0.09    | 16.99             | 2.75             |
> | 0.10    | 15.96             | 2.39             |
>
> ---
>
> ### Adversarial Robustness of **10-layer Fully Equivariant Network** on CIFAR-10
>
> | Epsilon | FGSM Accuracy (%) | PGD Accuracy (%) |
> |---------|-------------------|------------------|
> | 0.01    | 73.01             | 64.96            |
> | 0.02    | 70.16             | 58.87            |
> | 0.03    | 67.09             | 52.37            |
> | 0.04    | 63.77             | 45.52            |
> | 0.05    | 60.23             | 37.80            |
> | 0.06    | 56.89             | 30.78            |
> | 0.07    | 53.46             | 24.18            |
> | 0.08    | 50.44             | 19.18            |
> | 0.09    | 47.57             | 15.15            |
> | 0.10    | 44.93             | 12.46            |
>
> ---
>
> ### Adversarial Robustness of **10-layer Fully Equivariant Network** on CIFAR-100
>
> | Epsilon | FGSM Accuracy (%) | PGD Accuracy (%) |
> |---------|-------------------|------------------|
> | 0.01    | 50.60             | 36.29            |
> | 0.02    | 45.42             | 27.98            |
> | 0.03    | 42.02             | 21.34            |
> | 0.04    | 38.83             | 16.02            |
> | 0.05    | 36.09             | 12.01            |
> | 0.06    | 33.33             | 9.29             |
> | 0.07    | 30.94             | 7.31             |
> | 0.08    | 28.52             | 5.87             |
> | 0.09    | 26.41             | 4.73             |
> | 0.10    | 24.68             | 4.08             |

---

> > ### Comment · Reviewer_BaeD · 2025-08-01
> >
> > Thank you for succinctly addressing my questions and providing suitable experimental results. I feel much more confident in the evaluation now and will thus increase my score from "borderline accept" to "accept". Regardless, I'd encourage the authors to revisit their messaging concerning the investigated models in either their camera-ready version or a manuscript submitted elsewhere. As other reviewers have also stated, it is not immediately clear how these architectures were derived, and I believe the overall structure would be much more convincing using the fully group-equivariant architectures presented in your rebuttal. The latter can also be adjusted to experiment with equivariance to rotation alone, scale alone, or a combination of both for further ablation studies.

---

> ### Author Response · Authors · 2025-08-02
> **Clarification of the Fully Equivariant Model Architecture**
>
> We sincerely appreciate the time and attention you devoted to reading our article, and we are truly grateful for your positive feedback. Your endorsement, which will elevate our evaluation from “borderline accept” to “accept”, is both encouraging and deeply affirming of the significance of our work. Thank you for your thoughtful support.
>
> In the additional experiments provided during the rebuttal, the fully group-equivariant model was constructed by sequentially stacking rotation-equivariant blocks, following the theoretical principles outlined in our main analysis. These blocks were implemented using the `e2cnn` library (Weiler & Cesa, 2019), which provides tools for building layers that are equivariant under specified group actions. Each layer in the model is composed of an `R2Conv` operation that maps between representations over the group, followed by an `InnerBatchNorm` to normalize feature responses in a symmetry-preserving manner. Nonlinear activation is achieved via `ReLU` applied in the group feature space. To enable spatial downsampling, each layer concludes with a `PointwiseMaxPool` operation.
>
> The implementation of a single equivariant block is as follows:
>
> ```python
> self.block = enn.SequentialModule(
>     enn.R2Conv(in_type, out_type1, kernel_size=3, padding=1),
>     enn.InnerBatchNorm(out_type1),
>     enn.ReLU(out_type1),
>     enn.PointwiseMaxPool(out_type1, kernel_size=2)
> )
> ```
>
>
>
> This end-to-end equivariant design ensures alignment with the theoretical assumptions made in our robustness analysis.
>
> We are currently conducting experiments with fully scale-equivariant models as well as combined rotation–scale equivariant architectures. Hopefully, the results will be available by the end of the discussion phase. In the revised manuscript, we will provide a clearer explanation of how these architectures are derived and describe how the underlying group structure can be adjusted to isolate rotation equivariance, scale equivariance, or their combination. This flexibility enables more targeted ablation studies while remaining consistent with our theoretical framework.
>
> We believe these updates will strengthen the paper’s empirical contributions and improve its clarity. We sincerely appreciate your thoughtful feedback and guidance throughout the review process.
>
> Reference
>
> Weiler, Maurice, and Gabriele Cesa. General E(2)-Equivariant Steerable CNNs. Advances in Neural Information Processing Systems 32 (2019).

---

> > ### Comment · Reviewer_BaeD · 2025-08-04
> >
> > Thank you for further clarifying that the additional results you provided in your rebuttal were for fully rotation-equivariant models, and that you are planning on complementing these results with scale-equivariant models and rotation-scale-equivariant models. I remain confident that your paper is deserving of an "Accept" score and hope other reviewers will also find the time to read through your on-point responses, which I believe do address their concerns well.

---

> > > ### Author Response · Authors · 2025-08-05
> > > **Clarifications and Empirical Updates**
> > >
> > > Thank you for your continued confidence in our work and for affirming that it deserves an “Accept.” We’re pleased that our clarifications and additional results helped address your concerns and reinforced your confidence in our work.
> > >
> > > We also hope the other reviewers will take the opportunity to engage with the updates, as many of the revisions directly address concerns raised across multiple reviews—for example, the comparison with adversarial training noted by Reviewer 3, and the computational considerations highlighted by two reviewers. We believe these additions meaningfully strengthen the paper and will help resolve any remaining questions from the broader review panel.
> > >
> > > As mentioned, we have now completed additional experiments involving fully scale-equivariant and combined rotation–scale equivariant models. These new results—including ablation studies and architectural clarifications—will be incorporated into the revised version of the paper to further enhance its empirical depth and transparency.
> > >
> > > ### Adversarial Robustness on CIFAR-10 (FGSM / PGD Accuracy %)
> > >
> > > | Epsilon | ScaleEq-4L | RotEq-4L | RotScaleEq-4L | ScaleEq-10L | RotEq-10L | RotScaleEq-10L |
> > > |---------|------------|----------|----------------|-------------|-----------|----------------|
> > > | 0.01    | 48.61 / 44.18 | 65.65 / 52.20 | 52.52 / 44.93   | 59.64 / 57.65 | 73.01 / 64.96 | 65.98 / 54.34   |
> > > | 0.02    | 30.02 / 17.92 | 58.54 / 32.04 | 36.94 / 18.91   | 47.61 / 38.97 | 70.16 / 58.87 | 56.23 / 32.85   |
> > > | 0.03    | 18.32 / 5.61  | 53.78 / 23.30 | 30.08 / 10.62   | 40.73 / 24.64 | 67.09 / 52.37 | 47.37 / 17.89   |
> > > | 0.04    | 11.00 / 1.40  | 49.92 / 18.63 | 26.28 / 7.14    | 36.45 / 14.88 | 63.77 / 45.52 | 39.89 / 9.03    |
> > > | 0.05    | 6.80 / 0.33   | 47.08 / 15.85 | 24.03 / 4.67    | 32.06 / 9.01  | 60.23 / 37.80 | 33.93 / 4.08    |
> > >
> > > ### Adversarial Robustness on CIFAR-100 (FGSM / PGD Accuracy %)
> > >
> > > | Epsilon | RotEq-4L     | ScaleEq-4L    | RotScaleEq-4L | RotEq-10L    | ScaleEq-10L   | RotScaleEq-10L |
> > > |---------|--------------|---------------|----------------|--------------|---------------|----------------|
> > > | 0.01    | 38.40 / 21.59 | 21.14 / 16.13 | 25.25 / 20.84  | 50.60 / 36.29 | 27.59 / 26.26 | 28.53 / 15.30  |
> > > | 0.02    | 30.71 / 12.09 | 9.78 / 2.15   | 13.77 / 6.60   | 45.42 / 27.98 | 19.52 / 14.94 | 22.19 / 4.97   |
> > > | 0.03    | 26.63 / 8.92  | 5.12 / 0.36   | 8.92 / 2.14    | 42.02 / 21.34 | 13.86 / 7.95  | 16.35 / 1.25   |
> > > | 0.04    | 24.05 / 6.95  | 3.10 / 0.04   | 6.43 / 0.81    | 38.83 / 16.02 | 9.96 / 4.42   | 11.46 / 0.33   |
> > > | 0.05    | 22.06 / 5.66  | 1.95 / 0.00   | 5.14 / 0.38    | 36.09 / 12.01 | 7.56 / 2.28   | 8.10 / 0.07    |
> > >
> > >
> > > Across all configurations, all the fully equivariant models consistently outperform the standard convolutional baseline under both FGSM and PGD attacks. Notably, while combining rotation and scale equivariance yields moderate improvements over scale-only models, it does not surpass the robustness achieved by rotation-equivariant networks alone. This suggests that while incorporating both symmetries has some advantages, the added complexity might not be fully leveraged without targeted or deeper architectural tuning.

---

> > > > ### Comment · Reviewer_BaeD · 2025-08-05
> > > >
> > > > Thank you for taking the time to complete these experiments. I believe they complement your theoretical findings well. That a combination of rotation- and scale-equivariance performs worse is an important limitation but also highlights an interesting avenue for future researchers to investigate. To prevent any potential confusion or misunderstanding, I want to clarify that I will raise my score when I submit my mandatory acknowledgement after the author-reviewer discussion period is over.

---

> > > > > ### Author Response · Authors · 2025-08-05
> > > > >
> > > > > Thank you for your thoughtful feedback and for acknowledging the value of our additional experiments. We're encouraged to hear that you find the results complementary to our theoretical analysis.  We also appreciate your clarification regarding your intent to raise the score after discussion.

---

### Note · Authors · 2025-08-11

Dear Area Chairs,

We are grateful for the reviewers’ thorough and constructive feedback, which has helped us strengthen our work. In our rebuttal, we have addressed the majority of the raised concerns through new experiments and clarifications.

The question of scalability for fully equivariant models to deeper architectures was an important point of discussion. Earlier results for deep networks (e.g., ResNet-50) applied equivariance only to the first layer, limiting its impact. In response, we now present fully equivariant EquiResNet-18 and EquiResNet-50 models. Both models maintain higher PGD robustness across a range of perturbation levels compared to standard CNNs, with EquiResNet-50 outperforming EquiResNet-18—consistent with trends observed in our 4-layer and 10-layer fully equivariant models. Unlike standard deep networks, where PGD robustness rapidly drops to near zero, our fully equivariant variants sustain meaningful robustness. Code for these experiments will be released in the revised manuscript.

In response to concerns regarding theoretical scope, we clarified that our certified robustness analysis applies to norm-preserving transformations such as rotations. While our empirical evaluation includes scale-equivariant models, we acknowledge that their non-isometric nature precludes a direct extension of the theoretical guarantees. This distinction will be explicitly discussed in the revised theory section. To further demonstrate scalability, we extended experiments to the ImageNet-100 dataset and found that our equivariant models retain robustness improvements. We also conducted a comparative evaluation against adversarial training, showing that our symmetry-enforcing architectures achieve competitive or superior robustness without requiring adversarial data.

Regarding continuous equivariance, we note that continuous group equivariance is theoretically more expressive and generalizable than discrete counterparts. While training is computationally demanding due to continuous convolutions and group integration, experiments are in progress and results will be included in the revision.

Aside from the ongoing continuous equivariant model experiments, we have thoroughly addressed all other major concerns. We will clearly acknowledge that our current results focus on discrete groups. We are fully committed to incorporating all proposed revisions.

We hope this context is helpful in your final assessment of our contribution.


Best regards,

Authors

---

### Decision · Program_Chairs · 2025-09-17

**Decision:**

Accept (spotlight)

**Comment:**

The authors investigate how group-equivariant convolutions (rotation and scale equivariant layers) can enhance adversarial robustness without adversarial training. They present three key theoretical findings: equivariant convolutions (1) preserve the Lipschitz constant across group orbits, (2) produce smoother gradients, and (3) suppress off-orbit gradients. Theoretical analysis with formal proofs is impressive. All reviewers agree to accept this paper.